# VISUAL RECOGNITION WITH DEEP NEAREST CENTROIDS

**Wenguan Wang**[1*†]**, Cheng Han**[2*]**,Tianfei Zhou**[3*]**& Dongfang Liu**[2†]
CCAI, Zhejiang University[1], Rochester Institute of Technology[2] & ETH Zurich[3]

## ABSTRACT

We devise deep nearest centroids (DNC), a conceptually elegant yet surprisingly effective network for large-scale visual recognition, by revisiting Nearest Centroids, one of the most classic and simple classifiers. Current deep models learn the classifier in a *fully parametric* manner, ignoring the latent data structure and lacking explainability. DNC instead conducts nonparametric, case-based reasoning; it utilizes sub-centroids of training samples to describe class distributions and clearly explains the classification as the proximity of test data to the class sub-centroids in the feature space. Due to the distance-based nature, the network output dimensionality is flexible, and all the learnable parameters are *only* for data embedding. That means all the knowledge learnt for ImageNet classification can be *completely* transferred for pixel recognition learning, under the "pre-training and fine-tuning" paradigm. Apart from its nested simplicity and intuitive decision-making mechanism, DNC can even possess *ad-hoc* explainability when the sub-centroids are selected as actual training images that humans can view and inspect. Compared with parametric counterparts, DNC performs better on image classification (CIFAR-10, CIFAR-100, ImageNet) and greatly boosts pixel recognition (ADE20K, Cityscapes) with improved transparency, using various backbone network architectures (ResNet, Swin) and segmentation models (FCN, DeepLab$_{V3}$, Swin). Our code is available at DNC.

## 1 INTRODUCTION

Deep learning models, from convolutional networks (*e.g.*, VGG [1], ResNet [2]) to Transformer-based architectures (*e.g.*, Swin [3]), push forward the state-of-the-art on visual recognition. With these advancements, parametric softmax classifiers, which learn a set of parameters, *i.e.*, weight vector, and bias term, for each class, have become the *de facto* regime in the area (Fig. 1(b)). However, due to the parametric nature, they suffer from several limitations: **First**, they lack simplicity and explainability. The parameters in the classification layer are abstract and detached from the physical nature of the problem being modelled [4]. Thus these classifiers are hard to naturally lend to an explanation that humans are able to process [5]. **Second**, linear classifiers are typically trained to optimize classification accuracy only, paying less attention to modeling the latent data structure. For each class, only one single weight vector is learned in a fully parametric manner. Thus they essentially assume *unimodality* for each class [6, 7], less tolerant of intra-class variation. **Third**, as each class has its own set of parameters, deep parametric classifiers require the output space with a fixed dimensionality (equal to the number of classes) [8]. As a result, their transferability is limited; when using ImageNet-trained classifiers to initialize segmentation networks (*i.e.*, pixel classifiers), the last classification layer, whose parameters are valuable knowledge learnt from the image classification task, has to be thrown away.

In light of the foregoing discussions, we are motivated to present deep nearest centroids (DNC), a powerful, nonparametric classification network (Fig. 1(d)). Nearest Centroids, which has historical roots dating back to the dawn of artificial intelligence [9–14], is arguably the simplest classifier. Nearest Centroids operates on an intuitive principle: given a data sample, it is directly classified to the class of training examples whose mean (centroid) is closest to it. Apart from its internal transparency, Nearest Centroids is a classical form of exemplar-based reasoning [5, 11], which is fundamental to our most effective strategies for tactical decision-making [15] (Fig. 1(c)). Numerous past studies [16–18] have shown that humans learn to solve new problems by using past solutions of similar problems. Despite its conceptual simplicity, empirical evidence in cognitive science, and ever popularity [19–22], Nearest

---

*Equal contribution
†Corresponding author

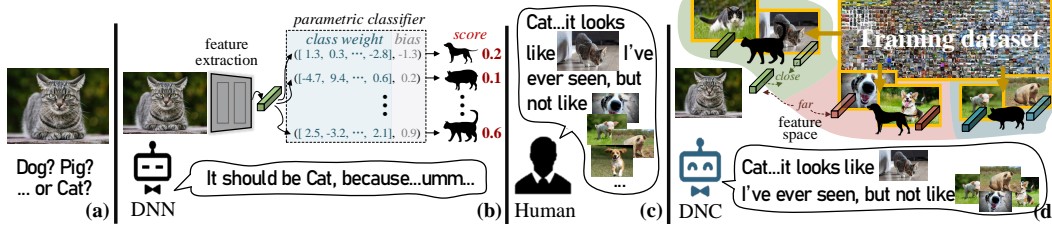

Figure 1: (b) Prevalent visual recognition models 🔲, built upon parametric softmax classifiers, have a few limitations, such as their non-transparent decision-making process. (c) Humans 👤 can use past cases as models when solving new problems [16, 18] (*e.g.*, comparing 🖼 with a few familiar/exemplar animals for categorization). (d) DNC 🔲 makes classification based on the similarity of 🖼 to class sub-centroids (representative training examples) in the feature space. The class sub-centroids are vital for capturing underlying data structure, enhancing interpretability, and boosting recognition.

Centroids and its utility in large datasets with high-dimensional input spaces are widely unknown or ignored by current community. Inheriting the intuitive power of Nearest Centroids, our DNC is able to serve as a *strong yet interpretable backbone* for large-scale visual recognition; it is fully aware of the aforementioned limitations of parametric counterparts while shows even better performance.

Specifically, DNC summarizes each class into a set of sub-centroids (sub-cluster centers) by clustering of training data inside the same class, and assigns each test sample to the class with the nearest sub-centroid. DNC is essentially an *experience-/distance*-based classifier – it merely relies on the proximity of test query to local means of training data ("quintessential" past observations) in the deep feature space. As such, DNC learns visual recognition by directly optimizing the representation, instead of deep parametric models needing an extra softmax classification layer after feature extraction. For training, DNC alternates between two steps: **i)** *class-wise clustering* for automatically discovering class sub-centroids, and **ii)** *classification prediction* for supervised representation learning, through retrieving the nearest sub-centroids. However, since the feature space evolves continually during training, computing the sub-centroids is expensive – it requires a pass over the full training dataset after each batch update and limits DNC's scalability. To solve this, we use a Sinkhorn Iteration [23] based clustering algorithm [24] for fast cluster assignment. We further adopt momentum update with an external memory for estimating online the sub-centroids (whose amount is more than 1K on ImageNet [25]) with small-batch size (*e.g.*, 256). Consequently, DNC can be efficiently trained by simultaneously conducting clustering and stochastic optimization on large datasets with small batches, only slowing the training speed slightly (*e.g.*, ∼5% on ImageNet).

DNC enjoys a few attractive qualities: **First**, improved *simplicity* and *transparency*. The intuitive working mechanism and statistical meaning of class sub-centroids make DNC elegant and easy to understand. **Second**, automated discovery of *underlying data structure*. By within-class deterministic clustering, the latent distribution of each class is automatically mined and fully captured as a set of representative local means. In contrast, parametric classifiers learn one single weight vector per class, intolerant of rich intra-class variations. **Third**, direct supervision of *representation learning*. DNC achieves classification by comparing data samples and class sub-centroids on the feature space. With such distance-based nature, DNC blends unsupervised sub-pattern mining (class-wise clustering) and supervised representation learning (nonparametric classification) in a synergy: local significant patterns are automatically mined to facilitate classification decision-making; the supervisory signal from classification directly optimizes the representation, which in turn boosts meaningful clustering. **Forth**, better *transferability*. DNC learns by *only* optimizing the feature representation, thus the output dimensionality no longer needs to be as many as the classes. With this algorithmic merit, all the useful knowledge (parameters) learnt from a source task (*e.g.*, ImageNet [25] classification) are stored in the representation space, and can be completely transferred to target tasks (*e.g.*, Cityscapes [26] segmentation). **Fifth**, *ad-hoc explainability*. If further restricting the class sub-centroids to be samples (images) of the training set, DNC can explain its prediction based on *IF · · · Then* rules and allow users to intuitively view the class representatives, and appreciate the similarity of test data to the representative images (detailed in §3&4.3). Such *ad-hoc* explainability [27] is valuable in safety-sensitive scenarios, and differs DNC from most existing network interpretation techniques [28–30] that only investigate *post-hoc* explanations and thus fail to elucidate precisely how a model works [31, 32].

DNC is an intuitive yet general classification framework; it is compatible with different visual recognition network architectures and tasks. We experimentally show: **In §4.1**, with ResNet [2] and Swin [3]

network architectures, DNC outperforms parametric counterparts on image classification, *i.e.*, **0.23-0.24%** `top-1` accuracy on CIFAR-10 [33] and **0.24-0.32%** on ImageNet [25], by *training from scratch*. **In §4.2**, when using our ImageNet-pretrained, nonparametric versions of ResNet and Swin as backbones, our pixel-wise DNC classifier greatly improves the segmentation performance of FCN [34], DeepLab$_{V_3}$ [35], and UperNet [36], on ADE20K [37] (**1.6-2.5%** `mIoU`) and Cityscapes [26] (**1.1-1.9%** `mIoU`). These results verify DNC's strong transferability and high versatility. **In §4.3**, after constraining class sub-centroids as training images of ImageNet, DNC becomes more interpretable, with only **0.12%** sacrifice in `top-1` accuracy (but is still **0.17%** better than the parametric counterpart).

These results are particularly impressive, considering the nonparametric and transparent nature of DNC. We feel this work brings fundamental insights into related fields.

## 2 RELATED WORK

**Distance-/Prototype-based Classifiers.** Among the numerous classification algorithms (*e.g.*, logistic regression [38], Naïve Bayes [39], random forest [40], support vector machines [41], and deep neural networks (DNNs) [42]), distance-based methods are particularly remarkable, due to their intuitive working mechanism. Distance-based classifiers are nonparametric and exemplar-driven, relying on similarities between samples and internally stored exemplars/prototypes. Thus they conduct case-based reasoning that humans use naturally in problem-solving, making them appealing and interpretable [16, 43]. $k$-Nearest Neighbors ($k$-NN) [9, 10] is a form of distance-based classifiers; it uses *all* training data as exemplars [44, 45]. Towards network implementation of $k$-NN [46–48], Wu *et al.* [49] made notable progress; their $k$-NN network outperforms parametric softmax based ResNet [2] and the learnt representation works well in few-shot settings. However, $k$-NN classifiers (including the deep learning analogues) cost huge storage space and pose heavy computation burden (*e.g.*, persistently retaining the training dataset and making full-dataset retrieval for each query) [50, 51], and the nearest neighbors may not be good class representatives [43]. Nearest Centroids [11–14] is another famous distance-based classifier yet has neither of the deficiencies of $k$-NN [20, 43]. Nearest Centroids selects representative class centers, instead of all the training data, as exemplars. Guerriero *et al.* [52] also investigate the idea of bringing Nearest Centroids into DNNs. However, they simply abstract each class into one single class mean, failing to capture complex class-wise distributions and showing weak results even in small datasets [33].

The idea of distance-based classification also stimulates the emergence of *prototypical networks*, which mainly focus on few-shot [53, 54] and zero-shot [55, 56] learning. However, they often associate to each class only one representation (prototype) [57] and their prototypes are usually flexible parameters [51, 53, 56] or defined prior to training [8, 55]. In DNC, a prototype (sub-centroid) is either a generalization of a number of observations or intuitively a typical training visual example. Via clustering based sub-class mining, DNC addresses two key properties of prototypical exemplars: *sparsity* and *expressivity* [58, 59]. In this way, the representation can be learnt to capture the underlying class structure, hence facilitating large-scale visual recognition while preserving transparency.

**Neural Network Interpretability.** As the *black-box* nature limits the adoption of DNNs in decision-critical tasks, there has been a recent surge of interest in DNNs' interpretability. However, most interpretation techniques only produce posteriori explanations for already-trained DNNs, typically by analysis of reverse-engineer importance values [28–30, 60–65] and sensitivities of inputs [66–69]. As many literature outlined, *post-hoc* explanations are problematic and misleading [32, 43, 70, 71]; they cannot explain what *actually* makes a DNN arrive at its decisions [72]. To pursue *ad-hoc* explainability, some attempts have been initiated to develop explainable DNNs, by deploying more interpretable machineries into black-box DNNs [73–75] or regularizing representation with certain properties (*e.g.*, sparsity [76], decomposability [77], monotonicity [78]) that can enhance interpretability.

DNC intrinsically relies on class sub-centroid retrieving. The theoretical simplicity makes it easy to understand; when anchoring the sub-centroids to available observations, DNC can derive intuitive explanations based on the similarities of test samples to representative observations. It simultaneously conducts representation learning and case-based reasoning, making it self-explainable without post-hoc analysis [27]. DNC relates to *concept*-based explainable networks [4, 5, 72–74, 79–81] that refer to human-friendly concepts/prototypes during decision making. These methods, however, necessitate nontrivial architectural modification and usually resort to pre-trained models, not to mention serving as backbone networks. In sharp contrast, DNC only brings minimal architectural change to parametric classifier based DNNs and yields remarkable performance on ImageNet [25] with training from scratch and *ad-hoc* explainability. It provides solid empirical evidence, for the first time as far as we know, for the power of case-based reasoning in large-scale visual recognition.

Figure 2: With a distance-/case-based classification scheme, DNC combines unsupervised sub-pattern discovery and supervised representation learning in a synergy.

## 3 DEEP NEAREST CENTROIDS (DNC)

**Problem Statement.** Consider the standard visual recognition setting. Let $\mathcal{X}$ be the visual space (*e.g.*, image space for recognition, pixel space for segmentation), and $\mathcal{Y} = \{1, \cdots, C\}$ the set of semantic classes. Given a training dataset $\{(x_n, y_n) \in \mathcal{X} \times \mathcal{Y}\}_{n=1}^{N}$, the goal is to use the $N$ training examples to fit a *model* (or hypothesis) $h : \mathcal{X} \mapsto \mathcal{Y}$ that accurately predicts the semantic classes for new visual samples.

**Parametric Softmax Classifier.** Current common practice is to implement $h$ as DNNs and decompose it as $h = l \circ f$. Here $f : \mathcal{X} \mapsto \mathcal{F}$ is a *feature extractor* (*e.g.*, convolution based or Transformer-like networks) that maps an input sample $x \in \mathcal{X}$ into a $d$-dimensional representation space $\mathcal{F} \in \mathbb{R}^d$, *i.e.*, $\boldsymbol{x} = f(x) \in \mathcal{F}$; and $l : \mathcal{F} \mapsto \mathcal{Y}$ is a *parametric classifier* (*e.g.*, the last fully-connected layer in recognition or last $1 \times 1$ convolution layer in segmentation) that takes $\boldsymbol{x}$ as input and produces class prediction $\hat{y} = l(\boldsymbol{x}) \in \mathcal{Y}$. Concretely, $l$ assigns a query $x \in \mathcal{X}$ to the class $\hat{y} \in \mathcal{Y}$ according to:

$$\hat{y} = \arg\max_{c \in \mathcal{Y}} s^c, \qquad s^c = (\boldsymbol{w}^c)^\top \boldsymbol{x} + b^c, \tag{1}$$

where $s^c \in \mathbb{R}$ indicates the unnormalized prediction score (*i.e.*, *logit*) for class $c$, $\boldsymbol{w}^c \in \mathbb{R}^d$ and $b^c \in \mathbb{R}$ are learnable parameters – class weight and bias term for $c$. Parameters of $l$ and $f$ are learnt by minimizing the softmax cross-entropy loss:

$$\mathcal{L} = \frac{1}{N} \sum_{n=1}^{N} - \log p(y_n | x_n), \quad p(y|x) = \mathrm{softmax}_y(l \circ f(x)) = \frac{\exp(s^y)}{\sum_{c \in \mathcal{Y}} \exp(s^c)}. \tag{2}$$

Though highly successful, the use of the parametric classifier $l$ has drawbacks as well: **i)** The weight matrix $\boldsymbol{W} = (\boldsymbol{w}^1, \cdots, \boldsymbol{w}^C) \in \mathbb{R}^{d \times C}$ and bias vector $\boldsymbol{b} = (b^1, \cdots, b^C) \in \mathbb{R}^d$ in $l$ are learnable parameters, which cannot provide any information about what makes the model $h$ reach its decisions. **ii)** $l$ makes the loss $\mathcal{L}$ only depend on the relative relation among logits, *i.e.*, $\{s^c\}_c$, and cannot directly supervise on the representation $\boldsymbol{x}$ [82, 83]. **iii)** $\boldsymbol{W}$ and $\boldsymbol{b}$ are learnt as flexible parameters, lacking explicit modeling of the underlying data structure. **iv)** The final output dimensionality is constrained to be the number of classes, *i.e.*, $C$. During transfer learning, as different visual recognition tasks typically have distinct semantic label spaces (with different number of classes), the classifier from a pretrained model has to be abandoned, even though the learnt parameters $\boldsymbol{W}$ and $\boldsymbol{b}$ are valuable knowledge from the source task.

The question naturally arises: might there be a simple way to address these limitations of current *de facto*, parametric classifier based visual recognition regime? Here we show that this is indeed possible, even with better performance.

**DNC Classifier.** Our DNC (Fig. 2) is built upon the intuitive idea of Nearest Centroids, *i.e.*, assign a sample $x$ to the class $\hat{y} \in \mathcal{Y}$ with the closest class center:

$$\hat{y} = \arg\min_{c \in \mathcal{Y}} \langle \boldsymbol{x}, \bar{\boldsymbol{x}}^c \rangle, \quad \bar{\boldsymbol{x}}^c = \frac{1}{N^c} \sum_{x_n^c : y_n^c = c} \boldsymbol{x}_n^c, \tag{3}$$

where $\langle \cdot, \cdot \rangle$ is a distance measure, given as: $\langle \boldsymbol{u}, \boldsymbol{v} \rangle = -\boldsymbol{u}^\top \boldsymbol{v} / \|\boldsymbol{u}\| \|\boldsymbol{v}\|$. For simplicity, all the features are defaulted to $\ell_2$-normalized from now on. $\bar{\boldsymbol{x}}^c$ is the mean vector of class $c$, $x_n^c$ is a training sample of $c$, *i.e.*, $y_n^c = c$, and $N^c$ is the number of training samples in $c$. As such, the feature-to-class mapping $\mathcal{F} \mapsto \mathcal{Y}$ is achieved in a *nonparametric* manner and *understandable* from user's view, in contrast to the parametric classifier $l$ that learns "non-transparent" parameters for each class. It makes more sense if multiple sub-centroids (local means) per class are used, which is in particular true for challenging visual recognition where complex intra-class variations cannot be simply described by the simple assumption of unimodality of data of each class. When representing each class $c$ as $K$ sub-centroids, denoted by $\{\boldsymbol{p}_k^c \in \mathbb{R}^d\}_{k=1}^{K}$, the $C$-way classification for sample $x$ takes place as a *winner-takes-all* rule:

$$\hat{y} = c^*, \quad (c^*, k^*) = \arg\min_{c \in \mathcal{Y}, k \in \{1, \cdots, K\}} \langle \boldsymbol{x}, \boldsymbol{p}_k^c \rangle. \tag{4}$$

Clearly, estimating class sub-centroids needs clustering of training samples within each class. As class

sub-centroids are sub-cluster centers in the latent feature space $\mathcal{F}$, they are locally significant visual patterns and can comprehensively represent class-level characteristics. DNC can be intuitively understood as selecting and storing prototypical exemplars for each class, and finding classification evidence for a previously unseen sample by retrieving the most similar exemplar. This also aligns with the *prototype theory* in psychology [17, 84, 85]: prototypes are a typical form of cognitive organisation of real world objects. DNC thus emulates the case-based reasoning process that we humans are accustomed to [27]. For instance, when ornithologists classify a bird, they will compare it with those typical exemplars from known bird species to decide which species the bird belongs to [43].

**Sub-centroid Estimation.** To find informative sub-centroids that best represent classes, we perform deterministic clustering within each class on the representation space $\mathcal{F}$. More specifically, for each class $c$, we cluster all the representations $\{\boldsymbol{x}_n^c \in \mathbb{R}^d\}_{n=1}^{N^c}$ into $K$ clusters whose centers are used as the sub-centroids of $c$, *i.e.*, $\{\boldsymbol{p}_k^c \in \mathbb{R}^d\}_{k=1}^K$. Let $\boldsymbol{X}^c = [\boldsymbol{x}_1^c, \cdots, \boldsymbol{x}_{N^c}^c] \in \mathbb{R}^{d \times N^c}$ and $\boldsymbol{P}^c = [\boldsymbol{p}_1^c, \cdots, \boldsymbol{p}_K^c] \in \mathbb{R}^{d \times K}$ denote the feature and sub-centroid matrixes, respectively. The deterministic clustering, *i.e.*, the mapping from $\boldsymbol{X}^c$ to $\boldsymbol{P}^c$, can be denoted as $\boldsymbol{Q}^c = [\boldsymbol{q}_1^c, \cdots, \boldsymbol{q}_{N^c}^c] \in \{0,1\}^{K \times N^c}$, where $n$-*th* column $\boldsymbol{q}_n^c \in \{0,1\}^K$ is an one-hot assignment vector of $n$-*th* sample $x_n^c$ *w.r.t* the $K$ clusters. $\boldsymbol{Q}^c$ is desired to maximize the similarity between $\boldsymbol{X}^c$ and $\boldsymbol{P}^c$, leading to the following binary integer program (BIP):

$$\max_{\boldsymbol{Q}^c \in \mathcal{Q}^c} \mathtt{Tr}\big((\boldsymbol{Q}^c)^\top (\boldsymbol{P}^c)^\top \boldsymbol{X}^c\big), \quad \mathcal{Q}^c = \{\boldsymbol{Q}^c \in \{0,1\}^{K \times N^c} | (\boldsymbol{Q}^c)^\top \mathbf{1}_K = \mathbf{1}_{N^c}\}, \tag{5}$$

where $\mathbf{1}_K$ is a $K$-dimensional all-ones vector. As in [24], we relax $\mathcal{Q}^c$ to be a *transportation polytope* [23]: $\mathcal{Q}'^c = \{\boldsymbol{Q}^c \in \mathbb{R}_+^{K \times N^c} | (\boldsymbol{Q}^c)^\top \mathbf{1}_K = \mathbf{1}_{N^c}, \boldsymbol{Q}^c \mathbf{1}_{N^c} = \frac{N^c}{K} \mathbf{1}_K\}$, casting BIP (5) into an *optimal transport* problem. In $\mathcal{Q}'^c$, besides the *one-hot assignment* constraint (*i.e.*, $(\boldsymbol{Q}^c)^\top \mathbf{1}_K = \mathbf{1}_{N^c}$), an *equipartition* constraint (*i.e.*, $\boldsymbol{Q}^c \mathbf{1}_{N^c} = \frac{N^c}{K} \mathbf{1}_K$) is added to inspire $N^c$ samples to be evenly assigned to $K$ clusters. This can efficiently avoid degeneracy, *i.e.*, mapping all the data to the same cluster. Then the solution can be given by a fast version [23] of Sinkhorn-Knopp algorithm [86], in a form of a normalized exponential matrix:

$$\boldsymbol{Q}^{c*} = \mathrm{diag}(\boldsymbol{\alpha}) \exp\big(\frac{(\boldsymbol{P}^c)^\top \boldsymbol{X}^c}{\varepsilon}\big) \mathrm{diag}(\boldsymbol{\beta}), \tag{6}$$

where the exponentiation is performed element-wise, $\boldsymbol{\alpha} \in \mathbb{R}^K$ and $\boldsymbol{\beta} \in \mathbb{R}^{N^c}$ are two renormalization vectors, which can be computed using a small number of matrix multiplications via Sinkhorn-Knopp Iteration [23], and $\varepsilon = 0.05$ trades off convergence speed with closeness to the original transport problem. In short, by mapping data samples into a few clusters under the constraints $\mathcal{Q}'^c$, we pursue *sparsity* and *expressivity* [58, 59], making class sub-centroids representative of the dataset.

**Training of DNC = Supervised Representation Learning + Automatic Sub-class Pattern Mining.** Ideally, according to class-wise cluster assignments $\{\boldsymbol{Q}^c\}_{c=1}^C$, we can get totally $CK$ sub-centroids $\{\boldsymbol{p}_k^c\}_{c,k=1}^{C,K}$, *i.e.*, mean feature vectors of the training data in the $CK$ clusters. Then the training target becomes as:

$$\mathcal{L} = \frac{1}{N} \sum_{n=1}^N -\log p(y_n|x_n), \quad p(y|x) = \frac{\exp\big(-\min(\{\langle \boldsymbol{x}, \boldsymbol{p}_k^y \rangle\}_{k=1}^K)\big)}{\sum_{c \in \mathcal{y}} \exp\big(-\min(\{\langle \boldsymbol{x}, \boldsymbol{p}_k^c \rangle\}_{k=1}^K)\big)}. \tag{7}$$

Comparing (2) and (7), as the class sub-centroids $\{\boldsymbol{p}_k^c\}_{c,k}$ are derived solely from data representations, DNC learns visual recognition by directly optimizing the representation $f$, instead of the parametric classifier $l$. Moreover, with such a nonparametric, distance-based scheme, DNC builds a closer link to metric learning [87–93]; DNC can even be viewed as learning a metric function $f$ to compare data samples $\{x_n\}_n$, under the guidance of the corresponding semantic labels $\{y_n\}_n$.

During training, DNC alternates two steps iteratively: **i)** class-wise clustering (5) for automatic sub-centroid discovery, and **ii)** sub-centroid based classification for supervised representation learning (7). Through clustering, DNC probes underlying data distribution of each class, and produces informative sub-centroids by aggregating statistics from data clusters. This automatic sub-class discovery process also enjoys a similar spirit of recent clustering based unsupervised representation learning [24, 94–101]. However, it works in a class-wise manner, since the class label is given. In this way, DNC optimizes the representation by adjusting the arrangement between sub-centroids and data samples. The enhanced representation in turn helps to find more informative sub-centroids, benefiting classification eventually. As such, DNC conducts *unsupervised sub-class pattern discovery* during *supervised representation learning*, distinguishing it from most (if not all) of current visual recognition models.

Since the latent representation $f$ evolves continually during training, class sub-centroids should be synchronized, which requires performing class-wise clustering over all training data after each batch

update. This is highly expensive on large datasets, even though Sinkhorn-Knopp iteration [23] based clustering (6) is highly efficient. To circumvent the expensive, offline sub-centroid estimation, we adopt *momentum update* and *online* clustering. Specifically, at each training iteration, we conduct class-wise clustering on the current batch and update each sub-centroid as:

$$\boldsymbol{p}_k^c \leftarrow \mu \boldsymbol{p}_k^c + (1 - \mu)\bar{\boldsymbol{x}}_k^c, \tag{8}$$

where $\mu \in [0, 1]$ is a momentum coefficient, and $\bar{\boldsymbol{x}}_k^c \in \mathbb{R}^d$ is the mean feature vector of data assigned to $(c, k)$-cluster in current batch. As such, the sub-centroids can be up-to-date with the change of parameters. Though effective enough in most cases, batch-wise clustering could not extend to a large number of classes, *e.g.*, when training on ImageNet [25] with 1K classes using batch size 256, not all the classes/clusters are present in a batch. To solve this, we store features from several prior batches in a memory, and do clustering on both the memory and current batch. DNC can be trained by gradient backpropagation in small-batch setting, with negligible lagging ($\sim$5% training delay on ImageNet).

**Versatility.** DNC is a general framework; it can be effortless integrated into any parametric classifier based DNNs, with minimal architecture change, *i.e.*, removing the parametric softmax layer. However, DNC changes the classification decision-making mode, reforms the training regime, and makes the reasoning process more transparent, without slowing the inference speed. DNC can be applied to various visual recognition tasks, including image classification (§4.1) and segmentation (§4.2).

**Transferability.** As a nonparametric scheme, DNC can handle an arbitrary number of classes with fixed output dimensionality ($d$); all the knowledge learnt on a source task (*e.g.*, ImageNet classification with 1K classes) are stored as a constant amount of parameters in $f$, and thus can be *completely* transferred for a new task (*e.g.*, Cityscapes [26] segmentation with 19 classes), under the "pre-training and fine-tuning" paradigm. In a similar setting, the parametric counterpart has to discard 2M parameters during transfer learning ($d$=2048 when using ResNet101 [2]). See §4.2 for related experiments.

**Ad-hoc Explainability.** DNC is a transparent classifier that has a built-in case-based reasoning process, as the sub-centroids are summarized from real observations and actually used during classification. So far we only discussed the case where the sub-centroids are considered as average mean feature vectors of a few training samples with similar patterns. When restricting the sub-centroids to be elements of the training set (*i.e.*, representative training images), DNC naturally comes with human-understandable explanations for each prediction, and the explanations are loyal to the internal decision mode and not created post-hoc. Studies regarding *ad-hoc* explainability are given in §4.3.

## 4 EXPERIMENT

### 4.1 EXPERIMENTS ON IMAGE CLASSIFICATION

**Dataset.** The evaluation for image classification is carried out on CIFAR-10[33] and ImageNet[25].

**Network Architecture.** For completeness, we craft DNC on popular CNN-based ResNet50/100 [2] and recent Transformer-based Swin-Small/-Base [3]. Note that, we only remove the last linear classification layer, and the final output dimensionality of DNC is as many as the last layer feature of the parametric counterpart, *i.e.*, 2,048 for ResNet50/100, 768 for Swin-Small, and 1,024 for Swin-Base.

**Training.** We use `mmclassification`[1] as codebase and follow the *default* training settings. For CIFAR-10, we train ResNet for 200 epochs, with batch size 128. The memory size for DNC models is set as 100 batches. For ImageNet, we train 100 and 300 epochs with batch size 16 for ResNet and Swin, respectively. The initial learning rates of ResNet and Swin are set as 0.1 and 0.0005, scheduled by a step policy and polynomial annealing policy, respectively. Limited by our GPU capacity, the memory sizes are set as 1,000 and 500 batches for DNC versions of ResNet and Swin, respectively. Other hyper-parameters are empirically set as: $K$=4 and $\mu$=0.999. Models are trained *from scratch* on eight V100 GPUs.

**Results on CIFAR-10.** Table 1 compares DNC with the parametric counterpart, based on the most representative CNN network architecture, *i.e.*, ResNet. As seen, DNC obtains better performance: DNC is **0.23**% higher on ResNet50, and **0.24**% higher on ResNet101, using fewer learnable parameters. With the exact same

Table 1: **Classification `top-1` accuracy** on CIFAR-10[33]`test`.#Params:the number of learnable parameters (same for other tables).

| Method | Backbone | #Params | top-1 |
|---|---|---|---|
| ResNet [2] | ResNet50 | 23.52M | 95.55% |
| **DNC**-ResNet | | 23.50M | **95.78**% |
| ResNet [2] | ResNet101 | 42.51M | 95.58% |
| **DNC**-ResNet | | 42.49M | **95.82**% |

backbone architectures and training settings, one can safely attribute the performance gain to DNC.

---

[1]https://github.com/open-mmlab/mmclassification

**Results on ImageNet.** Table 2 illustrates again our compelling results over different vision network architectures. In terms of `top-1` acc., our DNC exceeds the parametric classifier by **0.29**% and **0.28**% on ResNet50 and ResNet101, respectively. DNC also obtains promising results with Transformer architecture: **83.26**% *vs* 83.02% on Swin-S, **83.68**% *vs* 83.36% on Swin-B. These results are impressive, considering the transparent, case-based reasoning nature of DNC. One may notice another nonparametric, $k$-NN classi-

Table 2: **Classification `top-1` and `top-5` accuracy** on ImageNet [25] `val`.

| Method | Backbone | #Params | top-1 | top-5 |
|---|---|---|---|---|
| ResNet [2] | ResNet50 | 25.56M | 76.20% | 93.01% |
| **DNC**-ResNet | | 23.51M | **76.49**% | **93.08**% |
| ResNet [2] | ResNet101 | 44.55M | 77.52% | 93.06% |
| **DNC**-ResNet | | 42.50M | **77.80**% | **93.85**% |
| Swin [3] | Swin-S | 49.61M | 83.02% | 96.29% |
| **DNC**-Swin | | 48.84M | **83.26**% | **96.40**% |
| Swin [3] | Swin-B | 87.77M | 83.36% | 96.44% |
| **DNC**-Swin | | 86.75M | **83.68**% | **97.02**% |

fier [49] reports 76.57% `top-1` acc., based on ResNet50. However, [49] is trained with 130 epochs; in this setting, DNC gains 76.64%. Moreover, as mentioned in §2, [49] poses huge storage demand, *i.e.*, retaining the whole ImageNet training set (*i.e.*, 1.2M images) to perform the $k$-NN decision rule, and suffers from very low efficiency, caused by extensive comparisons between each test image and all the training images. These limitations prevent the adoption of [49] in real application scenarios. In contrast, DNC only relies on a small set of class representatives (*i.e.*, four sub-centroids per class) for classification decision-making and causes no extra computation budget during network deployment.

## 4.2 EXPERIMENTS ON SEMANTIC SEGMENTATION

**Dataset.** The evaluation for semantic segmentation is carried out on ADE20K [37] and Cityscapes [26].

**Segmentation Network Architecture.** For comprehensive evaluation, we approach DNC on three famous segmentation models (*i.e.*, FCN [34], DeepLab$_{V3}$[35], UperNet [36]), using two backbone architectures (*i.e.*, ResNet101 [2] and Swin-B [3]). For the segmentation models, the only architectural modification is the removal of the "segmentation head" (*i.e.*, 1×1 conv based, pixel-wise classification layer). For the backbone networks, we respectively adopt parametric classifier based and our nonparametric, DNC based versions, which are both trained on ImageNet [25] and reported in Table 2, for initialization. Thus for each segmentation model, we derive four variants from the different combinations of parametric and DNC versions of the backbone and segmentation network architectures.

**Training.** We adopt `mmsegmentation`[2] as the codebase, and follow the *default* training settings. We train FCN and DeepLab$_{V3}$ with ResNet101 using SGD optimizer with an initial learning rate 0.1, and UperNet with Swin-B using AdamW with an initial learning rate 6e-5. For all the models, the learning rate is scheduled following a polynomial annealing policy. As common practices [102, 103], we train the models on ADE20K `train` with crop size 512×512 and batch size 16; on Cityscapes `train` with crop size 769×769 and batch size 8. All the models are trained for 160K iterations on both datasets. Standard data augmentation techniques are used, including scale and color jittering, flipping, and cropping. The hyper-parameters of DNC are by default set as: $K = 10$ and $\mu = 0.999$.

**Performance on Segmentation.** As summarized in Table 3, our DNC segmentation models, no matter whether using DNC based backbones, obtain better performance over parametric competitors, *i.e.*, FCN + ResNet101, DeepLab$_{V3}$ + ResNet101, UperNet + Swin-B, across different segmentation and backbone network architectures. Taking as an example the famous FCN, the original version gains 39.91% and 75.6% `mIoU` on ADE20K and Cityscapes, respectively. By comparison, with the same backbone – ResNet101, DNC-FCN boosts the scores to 41.1%

Table 3: **Segmentation mIoU score** on ADE20K [37] `val` and Cityscapes [26] `val` (`top-1` acc. on ImageNet [25] `val` of backbones are also reported for reference).

| Method | Backbone | ImageNet top-1 acc. | #Params | ADE20K mIoU | Cityscapes mIoU |
|---|---|---|---|---|---|
| FCN [34] | ResNet101 [2] | 77.52% | 68.6M | 39.9% | 75.6% |
| | **DNC**-ResNet101 | 77.80% | 68.6M | 40.4%↑0.5 | 76.3%↑0.7 |
| **DNC**-FCN | ResNet101 [2] | 77.52% | 68.5M | 41.1%↑1.2 | 76.7%↑1.1 |
| | **DNC**-ResNet101 | 77.80% | 68.5M | **42.3**%↑2.4 | **77.5**%↑1.9 |
| DeepLab$_{V3}$ [35] | ResNet101 [2] | 77.52% | 62.7M | 44.1% | 78.1% |
| | **DNC**-ResNet101 | 77.80% | 62.7M | 44.6%↑0.5 | 78.7%↑0.6 |
| **DNC**-DeepLab$_{V3}$ | ResNet101 [2] | 77.52% | 62.6M | 45.0%↑0.9 | 79.1%↑1.0 |
| | **DNC**-ResNet101 | 77.80% | 62.6M | **45.7**%↑1.6 | **79.8**%↑1.7 |
| UperNet [36] | Swin-B [3] | 83.36% | 90.6M | 48.0% | 79.8% |
| | **DNC**-Swin-B | 83.68% | 90.6M | 48.4%↑0.4 | 80.1%↑0.3 |
| **DNC**-UperNet | Swin-B [3] | 83.36% | 90.5M | 48.6%↑0.6 | 80.5%↑0.7 |
| | **DNC**-Swin-B | 83.68% | 90.5M | **50.5**%↑2.5 | **80.9**%↑1.1 |

and 76.7%. When turning to DNC pre-trained backbone – DNC-ResNet101, our DNC-FCN outperforms again its parametric counterpart – FCN, *i.e.*, 42.3% *vs* 40.4% on ADE20K, 77.5% *vs* 76.3% on Cityscapes. Similar trends can be also observed for DeepLab$_{V3}$ and UperNet.

---

[2]https://github.com/open-mmlab/mmsegmentation

**Analysis on Transferability.** One appealing feature of DNC is its strong transferability, as DNC learns classification by directly comparing data samples in the feature space. The results in Table 3 also evidence this point. For example, DeepLab$_{V3}$ – even a parametric classifier based segmentation model – can achieve large performance gains, *i.e.*, 44.6% *vs* 44.1% on ADE20K, 78.7% *vs* 78.1% on Cityscapes, after using DNC-ResNet101 for fine-tuning. When it comes to DNC-DeepLab$_{V3}$, better performance can be achieved, after replacing ResNet101 with DNC-ResNet101, *i.e.*, 45.7% *vs* 45.0% on ADE20K, 79.8% *vs* 79.1% on Cityscapes. We speculate this is because, when the segmentation and backbone networks are both built upon DNC, the model only needs to adapt the original representation space to the target task, without learning any extra new parameters. Also, these impressive results reflect the innovative opportunity of applying our DNC for more downstream visual recognition tasks, either as a new classification network architecture or a strong and transferable backbone.

### 4.3 STUDY OF AD-HOC EXPLAINABILITY

So far, we have empirically showed that, by inheriting the intuitive power of Nearest Centroids and the strong representation learning ability of DNNs, DNC can serve as a transparent yet powerful tool for visual recognition tasks with improved transferability. When anchoring the class sub-centroids to real observations (*i.e.*, actual images from the training dataset), instead of selecting them as cluster centers (*i.e.*, mean features of a set of training images), one may expect that DNC will gain enhanced *ad-hoc* explainability. Next we will show this is indeed possible, only at negligible performance cost.

**Experimental Setup.** We still train DNC version of ResNet50 on ImageNet `train` for 100 epochs. In the first 90 epochs, the model is trained in a standard manner, *i.e.*, computing the class sub-centroids as cluster centers. Then we anchor each sub-centroid to its closest training image, based on the cosine similarity of features. In the final 10 epochs, the sub-centroids are only updated as the features of their anchored training images. Besides this, all the other training settings are as normal. In this way, we can get a more interpretable DNC-ResNet50.

**Interpretable Class Sub-centroids.** The top of Fig. 3 plots our discovered sub-centroid images for four

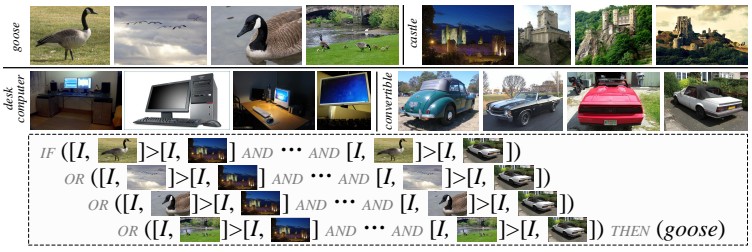

Figure 3: *Top:* sub-centroid images. *Bottom:* rule created for "goose".

ImageNet classes. These representative images are diverse in appearance, viewpoints, illuminations, scales, *etc.*, characterizing their corresponding classes and allowing humans to view and understand.

**Performance with Improved Interpretability.** We then report the score of our DNC-ResNet50 based on the interpretable class representatives on ImageNet `val`. As shown in Table 4, enforcing the class sub-centroids as real training images only brings marginal performance degradation (*e.g.*, 76.49%→ 76.37% `top-1` acc.), while coming with better interpretability. More impressively, ***our explainable DNC-ResNet50 even outperforms the vanilla black-box ResNet50***, *e.g.*, 76.37% *vs* 76.20%.

Table 4: **Classification `top-1` and `top-5` accuracy** on ImageNet [25] `val`, using cluster center *vs* resembling real observation as class sub-centroids, based on DNC-ResNet50 architecture.

| Sub-centroid | Architecture | `top-1` | `top-5` |
|---|---|---|---|
| *cluster center* | **DNC**-ResNet50 | 76.49% | 93.08% |
| *real observation* | | 76.37% | 93.04% |
| - | ResNet50[2] | 76.20% | 93.01% |

**Explain Inner Decision-Making Mode based on *IF ··· Then* Rules.** With the simple Nearest Centroids mechanism, we can use the representative images to form a set of *IF ··· Then* rules [4], so as to intuitively interpret the inner decision-making mode of DNC for human users. In particular, let $\hat{I}$ denote a sub-centroid image for class $c$, $\check{I}_{1:T}$ representative images for all the other classes, and $I$ a query image. One linguistic logical *IF ··· Then* rule can be generated for $\hat{I}$:

$$IF\ \big([I,\hat{I}] > [I,\check{I}_1]\ AND\ [I,\hat{I}] > [I,\check{I}_2]\ AND \cdots AND\ [I,\hat{I}] > [I,\check{I}_T]\big)\ THEN\ (\text{class } c), \qquad (9)$$

where $[\cdot,\cdot]$ stands for similarity, given by DNC. The final rule for class $c$ is created by combining all the rules of $K$ sub-centroid images $\hat{I}_{1:K}$ of class $c$ (see Fig. 3 bottom):

$$\begin{aligned}
IF\ &\big([I,\hat{I}_1] > [I,\check{I}_1]\ AND \cdots AND\ [I,\hat{I}_1] > [I,\check{I}_T]\big) \\
OR\ &\big([I,\hat{I}_2] > [I,\check{I}_1]\ AND \cdots AND\ [I,\hat{I}_2] > [I,\check{I}_T]\big) \\
OR\ &\cdots OR\ \big([I,\hat{I}_K] > [I,\check{I}_1]\ AND \cdots AND\ [I,\hat{I}_K] > [I,\check{I}_T]\big)\ THEN\ (\text{class } c).
\end{aligned} \qquad (10)$$

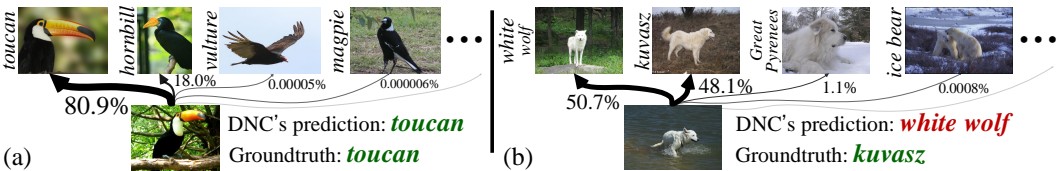

Figure 4: DNC can provide (dis)similarity-based interpretation. For the two test samples, we only plot the normalized similarities for their corresponding closest sub-centroids from top-4 scoring classes.

Table 5: A set of ablative experiments on ImageNet [25] `val` and ADE20K [37] `val`.

| $K$ | ImageNet | | $K$ | ADE20K |
| --- | --- | --- | --- | --- |
| | top-1 | top-5 | | mIoU |
| 1 | 77.31% | 93.01% | 1 | 43.2% |
| 2 | 77.54% | 93.32% | 5 | 44.0% |
| 3 | 77.68% | 93.63% | **10** | **44.3%** |
| **4** | **77.80%** | **93.85%** | 20 | 44.0% |

(a) Number of sub-centroids ($K$) for each class

| Memory (#*batch*) | ImageNet | |
| --- | --- | --- |
| | top-1 | top-5 |
| 0 | 77.49% | 93.09% |
| 700 | 77.58% | 93.16% |
| 800 | 77.64% | 93.35% |
| 900 | 77.75% | 93.67% |
| **1000** | **77.80%** | **93.85%** |

(b) Memory size

| $\mu$ | ImageNet | | ADE20K |
| --- | --- | --- | --- |
| | top-1 | top-5 | mIoU |
| 0 | 73.82% | 93.02% | 42.7% |
| 0.9 | 76.41% | 93.07% | 43.6% |
| 0.99 | 77.33% | 93.51% | 44.0% |
| **0.999** | **77.80%** | **93.85%** | **44.3%** |
| 0.9999 | 77.31% | 93.48% | 44.2% |

(c) Momentum coefficient ($\mu$)

**Interpret Prediction Based on (Dis)similarity to Sub-centroid Images.** Based on the interpretable class representatives, DNC can explain its predictions by letting users view and verify its computed (dis)similarity between query and class sub-centroid images. As shown in Fig. 4(a), an observation is correctly classified, as DNC thinks it looks (more) like a particular exemplar of "*toucan*". However, in Fig. 4(b), DNC struggles to assign the observation to two exemplars from "*white wolf*" and "*kuvasz*" respectively, and makes a wrong decision finally. Though users are unclear how DNC maps an image to feature, they can easily understand the decision-making mode [43] (*e.g.*, why is one class predicted over another), and verify the calculated (dis)similarity – the evidence for classification decision.

## 4.4 DIAGNOSTIC EXPERIMENT

To perform extensive ablation experiments, we train ResNet101 classification and DeepLab$_{V3}$ segmentation models for 100 epochs and 80K iterations, on ImageNet and ADE20K, respectively.

**Class Sub-centroids.** Table 5a studies the impact of the number of class sub-centroids ($K$) for each class. When $K = 1$, each class is represented by its centroid – the average feature vector of all the training samples of the class (Eq. 3), without clustering. The corresponding baseline obtains 77.31% `top-1` acc. and 43.2% `mIoU` for classification and segmentation, respectively. For classification, increasing $K$ from 1 to 4 leads to better performance (*i.e.*, 77.31% → 77.80%). This supports our hypothesis that one single class weight/center is far from enough to capture the underlying data distribution and proves the efficacy of our clustering based sub-class pattern mining. We stop using $K > 4$ as the required memory exceeds the computational limit of our hardware. We find similar trends on segmentation; using more sub-centroids ($K: 1 \to 10$) brings noticeable performance boost: 43.2% → 44.3%. However, increasing $K$ above 10 provides marginal or even negative gain. This may be because over-clustering finds some insignificant patterns, which are trivial or harmful for decision-making.

**External Memory.** We next study the influence of the external memory, only used in image classification. As shown in Table 5b, DNC gradually improves the performance as the increase of the memory size. It reaches 77.80% `top-1` acc. at size 1000. However, the results are still not reaching the performance saturating point, but rather the upper limit of our hardware's computational budget.

**Momentum Update.** Last, we ablate the effect of the momentum coefficient $\mu$ (Eq. 8) that controls the speed of sub-centroid online updating. From Table 5c we find the behaviors of $\mu$ are consistent in both tasks. In particular, DNC performs well with larger coefficients (*i.e.*, $\mu \in [0.999, 0.9999]$), signifying the importance of slow updating. The performance degrades when $\mu \in [0.9, 0.99]$, and encounters a large drop when $\mu = 0$ (*i.e.*, only using batch sub-centroids as approximations).

## 5 CONCLUSION

We present deep nearest centroids (DNC), building upon the classic idea of classifying data samples according to nonparametric class representatives. Compared to classic parametric models, DNC has merits in: **i)** systemic simplicity by bringing the intuitive Nearest Centroids mechanism to DNNs; **ii)** automated discovery of latent data structure using within-class clustering; **iii)** direct supervision of representation learning, boosted by unsupervised sub-pattern mining; **iv)** improved transferability that lossless transfers learnable knowledge across tasks; and **v)** *ad-hoc* explainability by anchoring class exemplars with real observations. Experiments confirm the efficacy and enhanced interpretability.

ACKNOWLEDGEMENT

This research was supported by the National Science Foundation under Grant No. 2242243.

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

## SUMMARY OF THE APPENDIX

This appendix contains additional details for the ICLR 2023 submission, titled *Visual Recognition with Deep Nearest Centroids"*. The appendix is organized as follows:

- §A provides the pseudo code of DNC.
- §B introduces more quantitative results on image classification.
- §C gathers additional semantic segmentation results on COCO-Stuff [104] dataset.
- §D presents the corresponding error bars on Tables 1, 2, and 3.
- §E investigates the potential of DNC in sub-categories discovery.
- §F reports the transferability of DNC towards other image classification task.
- §G evaluates the performance of DNC on ImageNetv2 test sets.
- §H compares the performance of using $k$-means and Sinkhorn-Knopp clustering algorithms.
- §I compares DNC with different distance (learning) based classifiers.
- §J reports additional diagnostic experiments for further investigations on $K$, update ratio of external memory, ImageNet capacity, feature size, and temperature parameter $\varepsilon$ in (6).
- §K offers more detailed discussions regarding the GPU memory cost.
- §L depicts more visual examples regarding the *ad-hoc* explainability of DNC.
- §M plots qualitative semantic segmentation results.
- §N gives additional review of representative literature on metric-/distance-learning and clustering-based unsupervised representation learning.
- §O discusses our limitations, societal impact, and directions of our future work.

## A   PSEUDO CODE OF DNC AND CODE RELEASE

The pseudo-code of DNC is given in Algorithm 1. To guarantee reproducibility, our code is available at https://github.com/ChengHan111/DNC.

**Algorithm 1** Pseudo-code of DNC in a PyTorch-like style.

```
# P: non-parametric sub-centroids (C x K x D)
# X: feature embeddings (N x D)

# C: number of classes
# K: number of sub-centroids for each class
# R: sinhorn-knopp iteration number
# mu: momentum coefficient (Eq.8)
# epsilon: hyper-parameter in the Sinkhorn-Knopp algorithm (Eq.6)

def DNC(X, label)
    #== Model Prediction and Training Loss (Eq.7) ==#

    # image-to-centroid assignment (N x K x C, Eq.5)
    L = torch.einsum('nd,ckd->nkc', X, P)
    output = torch.amax(L, dim=1)
    loss = CrossEntropyLoss(output, label)

    #======= Sub-centroid Estimation =======#
    for c in range(C)
        init_L = L[...,c]
        Q = online_clustering(init_L)

        # assignments and embeddings for images in class c
        m_c = L[label == c]
        x_c = X[label == c, ...]

        # find images that are assigned to each sub-centroid
        # and correctly classified
        m_c_tile = repeat(m_c, tile=K)
        m_q = Q * m_c_tile

        # find images with label c that are correctly classified
        x_c_tile = repeat(m_c, tile=x_c.shape[-1])
        x_c_q = x_c * x_c_tile
        f = torch.mm(m_q.transpose(), x_c_q)

        # num assignments for each sub-centroid of class c
        n = torch.sum(m_q, dim=0)

        # momentum update (Eq.8)
        if torch.sum(n) > 0:
            P_c = mu * P[c, n != 0, :] + (1-mu) * f[n != 0, :]
            P[c, n != 0, :] = P_c

    return loss

def online_clustering(L, iters=3, epsilon=0.05)
    Q = torch.exp(L / epsilon)
    Q /= torch.sum(Q)

    for _ in range(R):
        # row normalization
        Q /= torch.sum(Q, dim=1, keepdim=True)
        Q /= K

        # column normalization
        Q /= torch.sum(L, dim=0, keepdim=True)
        Q /= N

    # make sure the sum of each column to be 1
    Q *= N

    return one_hot(Q)
```

## B    MORE EXPERIMENTS ON IMAGE CLASSIFICATION

**Additional Results on CIFAR-10 [33].** CIFAR-10 dataset contains 60K (50K/10K for `train/test`) $32 \times 32$ colored images of 10 classes. Table 6 reports additional comparison results on CIFAR-10, based on ResNet-18 [2] network architecture. As seen, in terms of `top-1` accuracy, our DNC is **0.94**% higher than the parametric counterpart, under the same training setting.

Table 6: **Classification `top-1` accuracy** on CIFAR-10 [33] `test`. #Params: the number of learnable parameters (same for other tables). See §B for more details.

| Method | Backbone | #Params | top-1 |
|---|---|---|---|
| ResNet [2] | ResNet18 | 11.17M | 93.55% |
| **DNC**-ResNet | | 11.16M | **94.49**% |

**Additional Results on CIFAR-100 [33].** Table 7 reports comparison results on CIFAR-100, based on ResNet50 and ResNet101 network architectures [2]. CIFAR-100 dataset has 100 classes with 500 training images and 100 testing images per class. We can find that, DNC obtains consistently better performance, compared to the classic parametric counterpart. Specifically, DNC is **0.10**% higher on ResNet50, and **0.16**% higher on ResNet101, in terms of `top-1` accuracy.

Table 7: **Classification `top-1` accuracy** on CIFAR-100 [33] `test`, using lightweight backbone network architectures : MobileNet-V2 [105], and Swin-T [3]. See §B for more details.

| Method | Backbone | #Params | top-1 |
|---|---|---|---|
| ResNet [2] | ResNet50 | 23.71M | 79.81% |
| **DNC**-ResNet | | 23.50M | **79.91**% |
| ResNet [2] | ResNet101 | 42.70M | 79.83% |
| **DNC**-ResNet | | 42.49M | **79.99**% |

**Additional Results on ImageNet [25] using Lightweight Backbone Network Architectures.** Table 8 reports performance on ImageNet [25], using two lightweight backbone architectures: MobileNet-V2 [105], and Swin-T [3]. As can be seen, DNC, again, attributes decent performance. In particular, our DNC is **0.28**% and **0.30**% higher on MobileNet-V2 and Swin-Tiny, respectively. It is worth noticing that DNC can efficiently reduce the number of learnable parameters when the parametric classifier occupies a massive proportion in the original lightweight classification networks. Taking MobileNet-V2 as an example: our DNC reduces the number of learnable parameters from 3.50M to **2.22**M.

Table 8: **Classification `top-1` accuracy** on ImageNet [25] `val`. See §B for more details.

| Method | Backbone | #Params | top-1 |
|---|---|---|---|
| MobileNet [105] | MobileNet-V2 | 3.50M | 71.76% |
| **DNC**-MobileNet | | **2.22M** | **72.04**% |
| Swin [3] | Swin-T | 28.29M | 81.08% |
| **DNC**-Swin | | **27.52M** | **81.38**% |

## C   MORE EXPERIMENTS ON SEMANTIC SEGMENTATION

For through evaluation, we conduct extra experiments on COCO-Stuff [104], a famous semantic segmentation dataset. COCO-Stuff contains 9K/1K images for `train`/`test` of 80 object classes and 91 stuff classes. Similar to §4.2, we approach DNC on FCN [34], DeepLab$_{V3}$[35], and UperNet [36], using two backbone architectures, *i.e.*, ResNet101 [2] and Swin-B [3]. All models are obtained by following the standard training settings of COCO-Stuff `train`, *i.e.*, crop size $512\times512$, batch size 16, and 40K iterations. From Table 9 we can draw similar conclusions: in comparison with the parametric counterpart, DNC produces more precise segments and yields improved transferability.

Table 9: **Segmentation mIoU score** on COCO-Stuff [104]. See §C for more details.

| Method | Backbone | COCO-Stuff `test` mIoU |
|---|---|---|
| FCN [34] | ResNet101 [2] | 32.63% |
|  | **DNC**-ResNet101 | 32.89%↑**0.26** |
| **DNC**-FCN | ResNet101 [2] | 33.04%↑**0.41** |
|  | **DNC**-ResNet101 | **33.49**%↑**0.86** |
| DeepLab$_{V3}$ [35] | ResNet101 [2] | 36.01% |
|  | **DNC**-ResNet101 | 36.28%↑**0.27** |
| **DNC**-DeepLab$_{V3}$ | ResNet101 [2] | 36.51%↑**0.50** |
|  | **DNC**-ResNet101 | **36.79**%↑**0.78** |
| UperNet [36] | Swin-B [3] | 42.77% |
|  | **DNC**-Swin-B | 42.84%↑**0.07** |
| **DNC**-UperNet | Swin-B [3] | 43.13%↑**0.36** |
|  | **DNC**-Swin-B | **43.29**%↑**0.52** |

## D   ERROR BARS

In this section, we report standard deviation error bars on Tables 10, 11, and 12 for our main experiments regarding image classification (§4.1) and semantic segmentation (§4.2). The results are obtained by training the algorithm three times, with different initialization seeds.

Table 10: **Classification `top-1` accuracy** on CIFAR-10 [33] `test` with error bars. See §D for more details.

| Method | Backbone | #Params | top-1 |
|---|---|---|---|
| ResNet [2] | ResNet50 | 23.52M | 95.55 ± (0.14)% |
| **DNC**-ResNet |  | 23.50M | **95.78** ± (0.12)% |
| ResNet [2] | ResNet101 | 42.51M | 95.58 ± (0.13)% |
| **DNC**-ResNet |  | 42.49M | **95.82** ± (0.13)% |

Table 11: **Classification `top-1` and `top-5` accuracy** on ImageNet [25] `val` with error bars. See §D for more details.

| Method | Backbone | #Params | top-1 | top-5 |
|---|---|---|---|---|
| ResNet [2] | ResNet50 | 25.56M | 76.20 ± (0.10)% | 93.01% |
| **DNC**-ResNet |  | 23.51M | **76.49** ± (0.09)% | **93.08**% |
| ResNet [2] | ResNet101 | 44.55M | 77.52 ± (0.11)% | 93.06% |
| **DNC**-ResNet |  | 42.50M | **77.80** ± (0.10)% | **93.85**% |
| Swin [3] | Swin-S | 49.61M | 83.02 ± (0.14)% | 96.29% |
| **DNC**-Swin |  | 48.84M | **83.26** ± (0.13)% | **96.40**% |
| Swin [3] | Swin-B | 87.77M | 83.36 ± (0.12)% | 96.44% |
| **DNC**-Swin |  | 86.75M | **83.68** ± (0.12)% | **97.02**% |

Table 12: **Segmentation mIoU score** on ADE20K [37] `val` and Cityscapes [26] `val` with error bars. See §D for more details.

| Method | Backbone | ImageNet top-1 acc. | #Params | ADE20K mIoU | Cityscapes mIoU |
|---|---|---|---|---|---|
| FCN [34] | ResNet101 [2] | 77.52% | 68.6M | $39.9 \pm (0.11)\%$ | $75.6 \pm (0.13)\%$ |
| | **DNC**-ResNet101 | 77.80% | 68.6M | $40.4 \pm (0.11)\%$ | $76.3 \pm (0.12)\%$ |
| **DNC**-FCN | ResNet101 [2] | 77.52% | 68.5M | $41.1 \pm (0.10)\%$ | $76.7 \pm (0.11)\%$ |
| | **DNC**-ResNet101 | 77.80% | 68.5M | $\mathbf{42.3} \pm (0.10)\%$ | $\mathbf{77.5} \pm (0.11)\%$ |
| DeepLab$_{V3}$ [35] | ResNet101 [2] | 77.52% | 62.7M | $44.1 \pm (0.13)\%$ | $78.1 \pm (0.12)\%$ |
| | **DNC**-ResNet101 | 77.80% | 62.7M | $44.6 \pm (0.13)\%$ | $78.7 \pm (0.13)\%$ |
| **DNC**-DeepLab$_{V3}$ | ResNet101 [2] | 77.52% | 62.6M | $45.0 \pm (0.10)\%$ | $79.1 \pm (0.13)\%$ |
| | **DNC**-ResNet101 | 77.80% | 62.6M | $\mathbf{45.7} \pm (0.09)\%$ | $\mathbf{79.8} \pm (0.12)\%$ |
| UperNet [36] | Swin-B [3] | 83.36% | 90.6M | $48.0 \pm (0.11)\%$ | $79.8 \pm (0.13)\%$ |
| | **DNC**-Swin-B | 83.68% | 90.6M | $48.4 \pm (0.10)\ \%$ | $80.1 \pm (0.12)\%$ |
| **DNC**-UperNet | Swin-B [3] | 83.36% | 90.5M | $48.6 \pm (0.09)\ \%$ | $80.5 \pm (0.10)\%$ |
| | **DNC**-Swin-B | 83.68% | 90.5M | $\mathbf{50.5} \pm (0.09)\%$ | $\mathbf{80.9} \pm (0.10)\%$ |

# E   EXPERIMENTS ON SUB-CATEGORIES DISCOVERY

Through unsupervised, within-class clustering, our DNC represents each class as a set of automatically discovered class sub-centroids (*i.e.*, cluster center). This allows DNC to better describe the underlying, multimodal data structure and robusty depict for rich intra-class variance. In other words, our DNC can effectively capture sub-class patterns, which is conducive to algorithmic performance. Such a capacity of sub-patter mining is also considered crucial for good transferable features – representations learnt on coarse classes are capable of fine-grained recognition [106].

In order to quantify the ability of DNC for automatic sub-category discovery, we follow the experimental setup posed by [106] – learning the feature embedding using coarse-grained object labels, and evaluating the learned feature using fine-grained object labels. This evaluation strategy allows us to assess the feature learning performance regarding how well the deep model can discover variations within each category. A conjecture is that deep networks that perform well on this test have an exceptional capacity to identify and mine sub-class patterns during training, which the proposed DNC seeks to rigorously establish.

In particular, the network is first trained on coarse-grained labels with the baseline parametric softmax and with our non-parametric DNC using the same network architecture. After training on coarse classes, we use the top-1 nearest neighbor accuracy in the final feature space to measure the accuracy of identifying fine-grained classes. The classification performance evaluated in such setting is referred as **induction accuracy** as in [106]. Next we provide our experimental results on CIFAR100 [33] and ImageNet [25], respectively.

**Performance of Sub-category Discovery on CIFAR100.** CIFAR100 includes both fine-grained annotations in 100 classes and coarse-grained annotations in 20 classes. We examine sub-category discovery by transferring representation learned from 20 classes to 100 classes. As shown in Table 13, DNC consistently outperforms the parametric counterpart: DNC increases 0.12%, in terms of the standard `top-1` accuracy, on both ResNet50 and ResNet101 architectures. Nevertheless, when transferred to CIFAR100 (*i.e.*, 100 classes) using $k$-NN, a significant loss occurs on the baseline: 53.22% and 54.31% `top-1` acc. on ResNet50 and ResNet101, respectively. Our features, on the other hand, provide 14.24% and 13.82% improvement over the baseline, achieving 67.46% and 68.13% `top-1` acc. on 100 classes on ResNet50 and ResNet101, respectively. In addition, in comparison the parametric model, our approach results in only a smaller drop in transfer performance, *i.e.*, **-18.87** *vs* -32.99 on ResNet50, and **-18.47** *vs* -32.17 on ResNet101.

Table 13: **Top-1 induction accuracy** on CIFAR-100 [33] `test` using CIFAR-20 pre-trained models. Numbers reported with $k$-nearest neighbor classifiers. See §E for more details.

| Method | Backbone | 20 classes | 100 classes |
|---|---|---|---|
| ResNet [2] | ResNet50 | 86.21% | 53.22% |
| **DNC**-ResNet | | **86.33**% | **67.46**% |
| ResNet [2] | ResNet101 | 86.48% | 54.31% |
| **DNC**-ResNet | | **86.60**% | **68.13**% |

**Performance of Sub-category Discovery on ImageNet.** Table 14 provides experimental results of sub-category discovery on ImageNet `val`. As in [106], 127 coarse ImageNet categories are obtained by top-down clustering of 1K ImageNet categories on WordNet tree. Training on the 127 coarse classes, DNC improves the performance of baseline by 0.10% and 0.03%, achieving 84.39% and 85.91% on ResNet50 and ResNet101, respectively. When transferring to the 1K ImageNet classes using $k$-NN, our features provide huge improvements, *i.e.*, 8.98% and 9.29%, over the baseline.

Table 14: **Top-1 induction accuracy** on ImageNet [25] `val` using ImageNet-127 pre-trained models. Numbers reported with $k$-nearest neighbor classifiers. See §E for more details.

| Method | Backbone | 127 classes | 1000 classes |
|---|---|---|---|
| ResNet [2] | ResNet50 | 84.29% | 43.23% |
| **DNC**-ResNet | | 84.39% | **52.21**% |
| ResNet [2] | ResNet101 | 85.88% | 45.31% |
| **DNC**-ResNet | | 85.91% | **54.60**% |

The promising transfer results on CIFAR100 and ImageNet serve as strong evidence to suggest that our DNC is capable of automatically discovering meaningful sub-class patterns – latent visual structures that are not explicitly presented in the supervisory signal, and hence handle intra-class variance and boost visual recognition.

## F  TRANSFERABILITY TOWARDS OTHER IMAGE CLASSIFICATION TASK

In addition to conducting the coarse-to-fine transfer learning experiment (§E), we further evaluate the transfer learning performance by applying ImageNet-trained weight to Caltech-UCSD Birds-200-2011 (CUB-200-2011) dataset [107], following [108–110].

Specifically, CUB-200-2011 dataset comprises 11,788 bird photos arranged into 200 categories, with 5,994 for training and 5,794 for testing. All the models use ResNet50 architecture [2], and are trained for 100 epochs. SGD optimizer is adopted, where the learning rate is initialized as 0.01 and organized following a polynomial annealing policy. Standard data augmentation techniques are used, including flipping, cropping and normalizing. Experimental results are reported in Table 15. As seen, DNC is **+0.73**% and **+0.39**% higher in `Top-1` and `Top-5` acc., respectively. This clearly verifies that DNC owns better transferability.

Table 15: **Classification top-1 and top-5 accuracy** on CUB-200-2011 `test` [107]. See §F for more details.

| Model (ImageNet-trained) | Backbone | top-1 | top-5 |
|---|---|---|---|
| ResNet [2] | ResNet50 | 84.48% | 96.31% |
| **DNC**-ResNet | | **85.21**% | **96.70**% |

For parametric softmax classifier, both the feature network and softmax layer are fully learnable parameters. That means, for a new task, the softmax classifier has to both finetune the feature network and train a new softmax layer. However, for DNC, the feature network is the only learnable part. The class centers are not freely learnable parameters; they are directly computed from training data on the feature space. For a new task, DNC just needs to fine-tune the only learnable part – the feature network; the class centers are still directly computed from the training data according to the clustering assignments, without end-to-end training. Hence DNC owns better transferability during network fine-tuning.

## G    EVALUATION ON IMAGENETV2 TEST SETS

We evaluate DNC-ResNet50 on ImageNetv2 [111] test sets, *i.e.*, "Matched Frequency", "Threshold0.7" and "Top Images". Each test set contains 10 images for each ImageNet class, collected from MTurk. In particular, each MTurk worker is assigned with a certain classes. Then each worker is asked to select images belonging to his/her target class, from several candidate images sampled from a large image pool as well as ImageNet validation set. The output is a *selection frequency* for each image, *i.e.*, the fraction of MTurk workers selected the image in a task for its target class. Then three test sets are developed according to different principles defined on the selection frequency. For "Matched Frequency", [111] first approximated the selection frequency distribution for each class using those "re-annotated" ImageNet validation images. According to these class-specific distributions, ten test images are sampled from the candidate pool for each class. For "Threshold0.7", [111] sampled ten images from each class with selection frequency at least 0.7. For "Top Images", [111] selected the ten images with the highest selection frequency for each class. The results on these three test sets are shown in Table 16. As seen, our DNC exceeds the parametric softmax based ResNet50 by **+0.52-0.89**% `top-1` and **+0.26-0.47**% `top-5` acc..

Table 16: **Classification `top-1` and `top-5` accuracy** on ImageNetv2 `test` sets [111]. See §G for more details.

| test set | Method | Backbone | top-1 | top-5 |
|---|---|---|---|---|
| MatchedFrequency | ResNet [2] | ResNet50 | 63.30% | 84.70% |
|  | **DNC**-ResNet |  | **63.96**% | **85.17**% |
| Threshold0.7 | ResNet [2] | ResNet50 | 72.70% | 92.00% |
|  | **DNC**-ResNet |  | **73.59**% | **92.26**% |
| TopImages | ResNet [2] | ResNet50 | 78.10% | 94.70% |
|  | **DNC**-ResNet |  | **78.62**% | **94.96**% |

## H    SINKHORN-KNOPP *vs* $k$-MEANS CLUSTERING

To further probe the impact of Sinkhorn-Knopp based clustering [23], we further report the performance of DNC by using the classic k-means clustering algorithm, on CIFAR-10 and CIFAR100 datasets [33]. From Table 17 We can find that DNC with Sinkhorn-Knopp performs much better and is much more training-efficient.

Table 17: **Classification `top-1` and training time** on CIFAR-10 and CIFAR100 [33] `test` sets. See §H for more details.

| Dataset | Method | Backbone | top-1 | Training time (hours) |
|---|---|---|---|---|
| CIFAR10 [33] | **DNC**-$k$-means | ResNet50 | 93.88% | 4.5 |
|  | **DNC**-Sinkhorn |  | **95.78**% | **2.5** |
| CIFAR100 [33] | **DNC**-$k$-means | ResNet50 | 77.86% | 16.3 |
|  | **DNC**-Sinkhorn |  | **79.91**% | **3.7** |

# I   COMPARISON WITH DISTANCE (LEARNING) BASED CLASSIFIERS

We next compare our DNC with two metric based image classifiers [49, 52] and one distance learning based segmentation model [112].

We first compare DNC with DeepNCM [52]. DeepNCM conducts similarity-based classification using class means. As DeepNCM only describes its training procedures for CIFAR-10 and CIFAR-100 [33], we make the comparison on these two datasets to ensure fairness. From Table 18 we can find that, DNC significantly outperforms DeepNCM by **+2.11%** on CIFAR-10 and **+7.16%** on CIFAR-100, respectively.

DeepNCM simply abstracts each class into one single class mean, failing to capture complex within-class data distribution. In contrast, DNC considers $K$ sub-centers per class. Note that this is not just increasing the number of class representatives. This requires accurately discovering the underlying data structure. DNC therefore jointly conducts automated online clustering (for mining sub-class patterns) and supervised representation learning (for cluster center based classification). Finding meaningful class representatives is extremely challenging and crucial for Nearest Centroids. As the experiment in §H and Table 17 revealed, simply adopting classic $k$-means causes huge performance drop and significant training speed delay. Moreover, DeepNCM computes class mean on each batch, which makes poor approximation of the real class center. In contrast, DNC adopts the external memory for more accurate data densities modeling – DNC makes clustering over the large memory of numerous training samples, instead of the relatively small batch. Thus DNC better captures complex within-class variants and addresses two key properties of prototypical exemplars: sparsity and expressivity [58, 59], and eventually gains much more promising results.

Table 18: **Classification `top-1`** on CIFAR-10 and CIFAR100 [33] `test`. See §I for more details.

| Dataset | Method | Backbone | top-1 |
|---|---|---|---|
| CIFAR-10 | DeepNCM [52] 
 **DNC** | ResNet50 | 93.67% 
 **95.78**% |
| CIFAR-100 | DeepNCM [52] 
 **DNC** | ResNet50 | 72.75% 
 **79.91**% |

We also compare DNC with DeepNCA [49]. DeepNCA is a deep $k$-NN classifier, which conducts similarity-based classification based on top-$k$ nearest training samples. As [49] only reports results on ImageNet [25], we make the comparison on ImageNet to ensure fairness. Note that 130 training epochs are used, as in [49]. From Table 19 we can observe that, DNC outperforms DeepNCA. As we discussed in §2 and §4.1, DeepNCA poses huge storage demand, *i.e.*, retaining the whole ImageNet training set (*i.e.*, 1.2M images) to perform the $k$-NN decision rule, and suffers from very low efficiency, caused by extensive comparisons between each test sample and ALL the training images. These limitations prevent the adoption of [49] in real application scenarios. In contrast, DNC only relies on a small set of class representatives (*i.e.*, four sub-centroids per class) for decision-making and causes no extra computation budget during network deployment.

Table 19: **Classification `top-1`** on ImageNet [25] `val`. See §I for more details.

| Method | Backbone | top-1 |
|---|---|---|
| DeepNCA [49] 
 **DNC** | ResNet50 | 76.57% 
 **76.64**% |

Finally, we compare DNC with ContrastiveSeg [112] on Cityscapes [26] `val`, using ResNet101 [2] backbone and DeepLabV3 [35] segmentation architecture. ContrastiveSeg applies contrastive learning to better shape the feature embedding space, so as to improve semantic segmentation performance. However, ContrastiveSeg still relies on the softmax classifier; it is essentially a distance

learning boosted parametric classifier. From Table 19 we can observe that, DNC greatly surpasses ContrastiveSeg by 0.6% `mIoU`.

Table 20: **Segmentation mIoU score** on Cityscapes [26] `val`. See §I for more details.

| Method | Backbone | `mIoU` |
|---|---|---|
| ContrastiveSeg-DeepLab$_{V3}$ | ResNet101 | 79.1% |
| **DNC**-DeepLab$_{V3}$ | | **79.8%** |

These three experiments solidly demonstrate our effectiveness on both image classification and segmentation tasks, even compared with other distance (learning) based counterparts.

## J  ADDITIONAL DIAGNOSTIC EXPERIMENT

**Output Dimensionality.** As stated in §4, the final output dimensionality of our DNC is set as many as the one of the last layer of the parametric counterpart, for the sake of fair comparison. However, owing to the distance-/similarity-based nature, DNC has the flexibility to handle any output dimensionality. In Table 21, we further study the influence of the output dimensionality of DNC. As seen, when setting the final output dimensionality as 1280, we can achieve **76.61%** `top-1` acc., which is higher than the initial 2048 dimension configuration, *i.e.*, 76.49%. We attribute the reason for the better balance between memory capacity and feature dimensionality with the limitation of hardware computational budget – when reducing the final output dimensionality, the expressibility of the final feature is weakened but more image features can be stored in the external memory for more accurate sub-centroid estimation.

Table 21: **Ablative experiments regarding the final output dimensionality** on ImageNet [25] `val`. See §J for more details.

| Output | ImageNet | |
| dimensionality | `top-1` | `top-5` |
|---|---|---|
| 640 | 76.23% | 92.83% |
| 1024 | 76.28% | 92.90% |
| **1280** | **76.61**% | **93.12**% |
| 2048 | 76.49% | 93.08% |

**Temperature Parameter** $\varepsilon$ **in (6).** Parameter $\varepsilon$ in (6) trades off convergence speed with closeness to the original transport problem [23, 24]. In Table 22, we further study the impact of $\varepsilon$ on ImageNet [25] `val`.

Table 22: **Ablative experiments regarding temperature parameter** $\varepsilon$ **in (6)** on ImageNet [25] `val`. See §J for more details.

| $\epsilon$ | ImageNet | |
| | `top-1` | `top-5` |
|---|---|---|
| 0.01 | 76.34% | 92.97% |
| **0.05** | **76.49**% | **93.08**% |
| 0.1 | 76.40% | 93.02% |

**Number of Centroids** $K$**.** As shown in Table 23, we set $K$ with different values based on the number of training samples of each class. Specifically, ImageNet contains between 732 and 1300 training images (#images) per class. Then, $K = 1$ is assigned to the class having between 732 and 874 training samples, $K = 2$ to the class having between 875 and 1016 samples, $K = 3$ to the class

having between 1017 and 1158 samples, and $K = 4$ to the class having between 1159 and 1300 samples. We can find that we gain slightly better performance, +0.06% higher in `top-1` acc. when compared with fixing $K = 4$ for all the classes.

Table 23: **Ablative experiments with varying $K$ on different classes** on ImageNet [25] `val`. See §J for more details.

| $K$ | ImageNet | |
|---|---|---|
| range | `top-1` | `top-5` |
| unique value 4 | 76.49% | 93.08% |
| varying between 1 and 4 | **76.55**% | **93.10**% |

## K  MEMORY COST

In our experiment, we only adopt external memory for ImageNet classification. Below we provide more discussion regarding this point. Semantic segmentation is a pixel-wise classification task, where each training image provides numerous pixel samples for each class. For ImageNet classification, however, each training image is only assigned to one single class. Moreover, ImageNet has 1K classes, while in general semantic segmentation only has dozens of classes. Therefore, for a training mini-batch of, for example, 256 images, every class in segmentation usually has many training pixel samples in each mini-batch; this allows us to use a large $K$ for clustering. However, under the same setting, for each mini-batch, there must have many ImageNet classes that do not have corresponding image samples – we have 1000 classes but each mini-batch only has 256 training images. This is why we need to build an external memory during ImageNet classification. This is also why applying Nearest Centroids for batch-wise ImageNet classification training is extremely challenging; [52] cannot handle ImageNet classification, as it only computes class means in a batch-wise manner.

Table 24a and 24b provide statistics of GPU memory cost (per GPU usage) with respect to the number of class centroids $K$ and external memory size respectively. The statistics are gathered during the training of DNC-ResNet50 on ImageNet, using eight V100 GPUs. In our experiment, we set the size of the external memory as 256,000 image samples (*i.e.*, 1000 batches) and $K = 4$ for DNC-ResNet50 (*i.e.*, a total of 4000 class sub-centroids on ImageNet). More specifically, for a memory with #batch=1000, it stores 8 #gpu $\times$ 32 #batch size $\times$ 1000 #batch = 256000 #image examples. In comparison, for segmentation, when we set $K = 10$ (*i.e.*, a total of 1500 class sub-centroids for ADE20K), we have 65,536 pixel training samples in each mini-batch of 16 training images, without using memory.

Table 24: **Statistics of GPU memory cost** with respect to the number of class centroids $K$ and external memory size, where 1000 batches = 256000 image samples. See §K for more details.

| $K$ (Fixed memory size: 1000 batches) | GPU memory cost (GB per GPU) |
|---|---|
| 1 | 16.07 |
| 2 | 19.04 |
| 3 | 22.03 |
| 4 | 26.31 |

(a) GPU memory cost *w.r.t* number of centroids $K$

| memory size (Fixed $K = 4$) | GPU memory cost (GB per GPU) |
|---|---|
| 400 batches | 11.88 |
| 600 batches | 16.35 |
| 800 batches | 21.65 |
| 1000 batches | 26.35 |

(b) GPU memory cost *w.r.t* external memory size

## L  ADDITIONAL STUDY OF AD-HOC EXPLAINABILITY

**Interpretable Class Sub-centroids.** In Fig. 5, we show more examples of sub-centroid images for eight ImageNet classes. These representative images are automatically discovered by DNC, and can be intuitively viewed by users. As seen, the class sub-centroid images are able to capture diverse characteristics of their classes, in the aspects of appearance, viewpoints, scales, illuminations, *etc*.

**Interpret Prediction Based on (Dis)similarity to Sub-centroid Images.** Fig. 6 provides more results regarding the (dis)similarity-based interpretation of DNC prediction. As seen, based on the similarity of test images to the class sub-centroid images, users can clearly understand the decision making mode and make verification. DNC's compelling explainability enables it to establish trustworthiness with humans and empowers its potential in high stake applications.

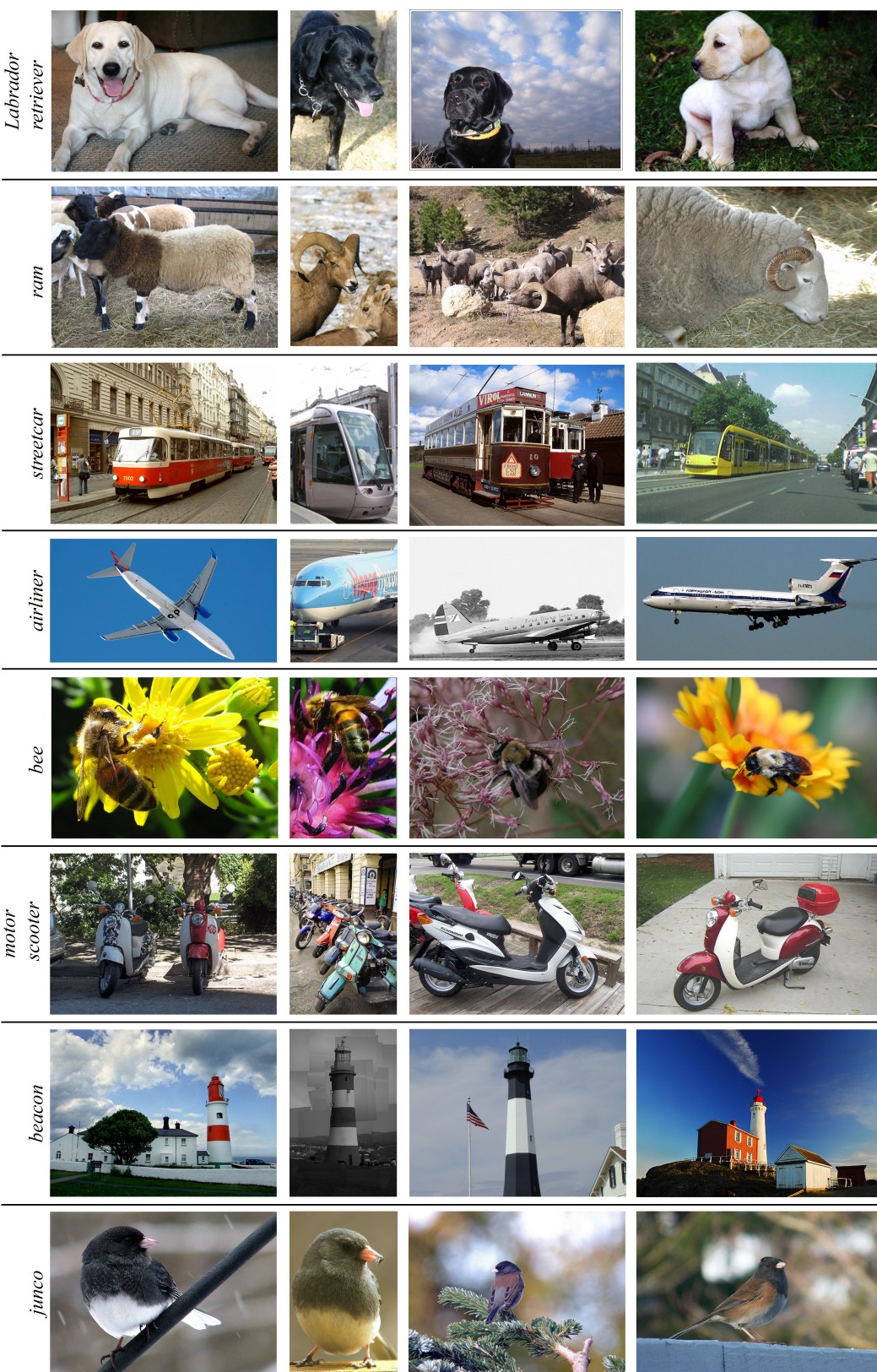

Figure 5: Sub-centroid images for eight randomly chosen classes from ImageNet [25]. See §L for more details.

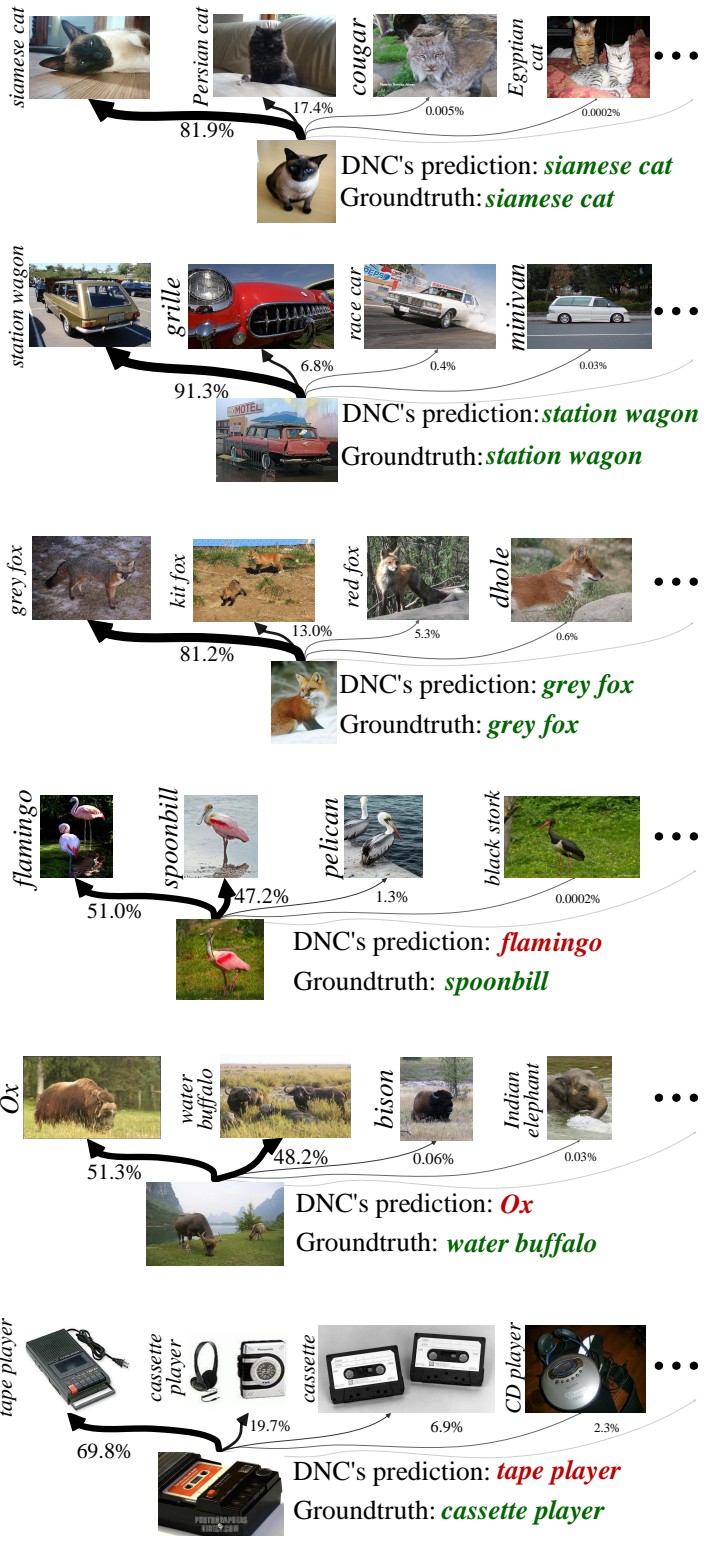

Figure 6: More examples on DNC interpreting its predictions based on its computed similarity to class sub-centroid images. For each test image, we plot the normalized similarities for the corresponding closest sub-centroids from the top-4 scoring classes. See §L for more details.

## M    QUALITATIVE RESULTS ON SEMANTIC SEGMENTATION

Fig. 7 and Fig. 8 illustrate a few representative visual examples of semantic segmentation results on ADE20K [37] and Cityscapes [26], respectively. In comparison with the parametric counterpart, our approach produces more precise segments in different challenging scenes (for example, where objects with drastic photometric or geometric appearances). For instance, UperNet [36]-Swin-B [3] confuses on neighbouring objects with similar colors (*e.g.*, desk and chair) and leaves a large false-negative regions (see the first image of Fig. 7); it also has difficulties in segmenting small scale objects (*e.g.*, motorbike semantic parsing). Among these examples, DNC consistently demonstrates supreme performance. Essentially, we argue that the proposed DNC has a stronger ability to supervises the pixel embedding space via anchoring sub-centroids directly, leading to a better predictions on segmenting such hard cases.

## N    ADDITIONAL LITERATURE REVIEW

This section gives additional review of representative literature on metric-/distance-learning and clustering-based unsupervised representation learning.

**Metric Learning.** The goal of metric learning (also a.k.a distance learning) is to learn a distance metric/embedding which brings together similar samples and pushes away dissimilar ones. Metric learning has a long history, dating back to some early work for more than few decades ago [113, 114]. In particular, diverse metric learning objective functions, such as contrastive loss [87, 93, 115], triplet loss [116], quadruplet loss [117], and $n$-pair loss [88], were proposed to measure similarity in the feature space for representation learning, and showed significant benefit in a wide range of applications, such as image retrieval [118], face recognition [116, 119–122], and person re-identification [123], to name a few representative ones. Recently, metric learning gained astonishing success in learning transferable deep representations from massive unlabeled data [89]. A family of *instance-based* approaches used the contrastive loss [124, 125] to explicitly compare pairs of image representations [125–129]. Another group of methods adopted a *clustering-based* strategy; they learn unsupervised representations by discriminating between groups of images without expensive pair-wise comparison between image instances [24, 94–101]. More recently, there are some efforts that revisit the idea of metric learning in supervised learning setting [91–93, 112].

As distance-/similarity-based classifiers rely on the similarity between samples and class representatives for classification, the fields of metric learning and distance-based classification are naturally related and the selection of a proper distance measure impacts the success of distance-based classifiers [130]. Historically, metric learning and class center discovery are two critical research topics in the field of distance-based classification. As a nonparametric, distance-based classifier, DNC can be viewed as a learnable metric function, which is trained to compare data samples under the guidance of the corresponding semantic labels. Although current distance learning based algorithms also optimize the feature space by comparing data samples, they need parametric softmax for classification. Their trained models are still black-box parametric classifiers without any interpretability. In sharp contrast, DNC directly assigns an observation to the class of the closet centroids, without using parametric softmax. Moreover, its distance-based classification decision-making mode allows DNC to effortless adopt existing metric learning techniques (and the way of its current training can be already viewed as performing metric learning).

**Clustering-based Self-supervised Representation Learning.** There is a recent trend to bind self-supervised representation learning with clustering. Basically, clustering-based self-supervised representation learning is more efficient for large-scale training data and more tolerant of the similarity (semantic structure) among data samples, compared with the instance-level counterpart. More specifically, early approaches [24, 94, 96, 97, 100, 131, 132] learn representations of image samples and cluster assignments in an *alternative* manner, *i.e.*, group features into clusters to derive pseudo supervisory signal and subsequently employ it for supervising representation learning. In very recent, numerous efforts have been devoted to *simultaneous* clustering and representation learning based on, *e.g.*, data reconstruction [95, 133], mutual information maximization [131, 134, 135], or contrastive instance discrimination [98, 99, 101, 136–138].

Our work is also related to these clustering based unsupervised representation learning methods, especially the ones [24, 98] resorting to the fast Sinkhorn-Knopp algorithm [23] for robust clus-

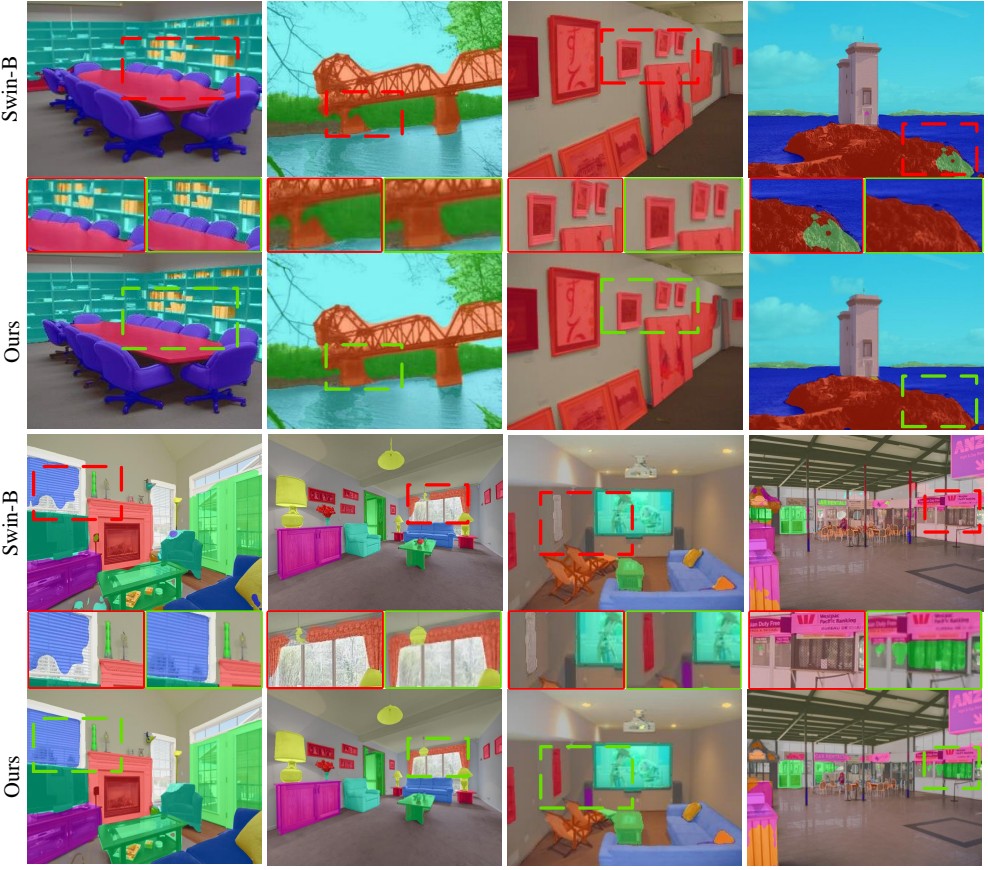

Figure 7: **Qualitative semantic segmentation results** of UperNet [36]-Swin-B [3] and DNC on ADE20K [37] val. Red and green bounding boxes represent the same zoom-in area on UperNet [36]-Swin-B [3] and DNC, respectively. See §M for more details.

tering. They aim to *learn transferable representation* from massive *unlabeled* data. Although also involving a similar clustering procedure for automatic sub-pattern mining, DNC targets at building a strong *similarity-based classification network* in the standard *supervised learning* setting. In DNC, the automatically discovered class sub-centroids are informative class representatives, which explicitly capture latent data structure of each class, and serve as classification evidence with clear physical meaning. The whole training procedure is a hybrid of class-wise online clustering (for unsupervised sub-class discovery) and sub-centroid based classification (for supervised representation learning). This well addresses the nature of Nearest Centroids and brings novel insights into the visual recognition task itself.

## O    LIMITATION AND FUTURE WORK

**Limitation.** One limitation of our approach is that the Sinkhorn-Knopp algorithm runs in time $\widetilde{O}(\frac{n^2}{\epsilon^3})$ which would reduce the training efficiency. Though in practice, we find 3 sinkhorn loops per training iteration is sufficient enough for model representation, bringing a minor computational overhead (*i.e.*, ∼5% training delay on ImageNet). This also indicates possible directions for our future research.

**Social Impact.** This work introduces DNC possessing the nature of nested simplicity, intuitive decision-making mechanism and even *ad-hoc* explainability. On positive side, the approach advances model accuracy and is valuable in safety-sensitive applications by showing the advanced robustness on sub-categories discovery, *e.g.*, quality analytics, autonomous driving [139–141], *etc.*. For potential negative social impact, our DNC struggles in handling out-of-distribution data, which

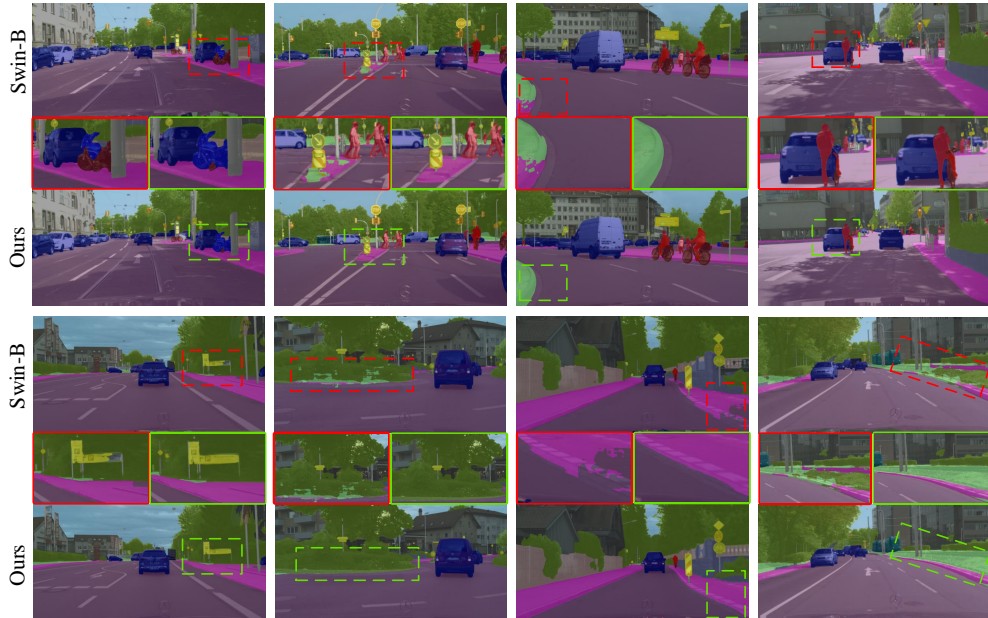

Figure 8: **Qualitative semantic segmentation results** of UperNet [36]-Swin-B [3] and DNC on Cityscapes [26] val. Red and green bounding boxes represent the same zoom-in area on UperNet [36]-Swin-B [3] and DNC, respectively. See §M for more details.

is a common limitation of all the discriminative classifiers. Hence its utility in open-world scenarios should be further examined.

**Future Work.** Despite DNC's systemic simplicity and efficacy, it also comes with new challenges and unveils some intriguing questions. For example, incorporating more powerful, time-efficient online clustering algorithms into DNC might improve training speed and test accuracy. Also, the number of class centroids $K$ currently is set to a fixed value for all classes, which may not be optimal given that intra-class variability varies across classes. Our experiments in §J and Table 23 also suggest that simply varying $K$ with the number of training samples of the class can boost performance. Thus adopting the clustering algorithms that do not require a predefined and fixed number of clusters [142] may allow DNC to automatically determine $K$ for different classes, which eventually benefit performance. In addition, instead of only considering first-order statistics, DNC could be enhanced by second-order statistics, which contain more useful information, but must contend with the computational overhead they impose. Another essential future direction deserving of further investigation is the in-depth analysis of the intrinsic properties of DNC, such as its robustness against perturbation, adversarial attack [143, 144], and out-of-distribution data, with the comparison of the softmax based counterpart. This endeavor would help us to better understand the nature of parametric and nonparametric classifiers and reveals directions for further improvement. Furthermore, we will explore the possibility of unifying close-set and open-world visual recognition within our framework. Finally, considering the similarity-/distance-based nature of DNC, the incorporation of metric learning based training objectives is also another promising direction for further boosting the performance. Given the vast number of technique breakthroughs in recent years, we expect a flurry of innovation towards these promising directions. Overall, we believe the results presented in this paper warrant further exploration.

