# OpenReview forum: "Visual Recognition with Deep Nearest Centroids"
_ICLR.cc/2023/Conference — ICLR 2023 notable top 25%_

### Official Review · Reviewer_k9FP · 2022-10-18

**Confidence:** 4
**Correctness:** 4
**Technical Novelty And Significance:** 3
**Empirical Novelty And Significance:** 3
**Recommendation:** 8

**Clarity, Quality, Novelty And Reproducibility:**

**Novelty**

The proposed deep centroids layer can be considered as the application of the same algorithm for cluster assignments done in unsupervised learning [1, 2] to do within class cluster assignments. The authors should be upfront about this. Note that this does not mean, that the current work is not novel, but the writing in the introduction/related work should reflect this and the work should be placed in this context.

**Reproducibility**

* It would be nice to have a small Algorithm section showing the pseudcode of the algorithm. For example, the algorithm can show in what sequence the cluster assignments, (parameter, memory bank and cluster) updates are done. For reference, look at Algorithm 1 of the MoCo paper (https://arxiv.org/abs/1911.05371)
* The deep centroids layer has 2 additional hyperparameters (K and $\mu$). How was hyperparameter tuning done? Was there a separate validation set split from the train set? Please provide these details.

**Clarity and Quality**

I would appreciate it if the authors can clarify these details.

* **Section 3, transferrability**: The authors claim that for after pretraining, the softmax layer is thrown away, leading to "wastage" of parameters. This is not 100% convincing since the equivalence to the deep nearest centroids is the "sub-centroids" which are K times more than the paramters in the softmax layer which also have to be thrown away during finetuneing.
* How important is the Sinkhorn-Knapp algorithm to determine sub-centroid assignments? Can one just set the cluster assignment to be the nearest sub-centroid as done in K-Means?
* **Section 4.3**: For the "anchoring of sub-centroids", to obtain the images used to anchor the sub-centroids, do the authors find the closest image over the entire training set or is just the memory bank sufficient?
* **Parametric Softmax Classifier**: "and cannot directly supervise on the representation $x$". It is unclear what this means as $s$ is a a function of $x$
* **Memory Bank Size**: It might be more intutive to show this hyperparameter as a function of #examples rather than #batches.
* W and b are learnt without considering any underlying data distribution. It is unclear what this means as W and b are learnt by modeling the  conditional P(y|x) which is the data distribution.


The paper introduces some terms which in my opinion are a bit colloquial and diffficult to understand. The colloquail terms are bolded below

* Abstract: Case-based reasoning, it is unclear what "case-based" refers to
* Intro, para 1: "detached from the physical nature of the problem"
* Intro, para 2: it is fully aware of the limitations of parametric counterparts
* nonparametric manner and understandable from user’s view, in contrast to the parametric classifier that learns **non-transparent parameters** for each class.
* Section DNC Classifier: finding **classification evidence** for a previously unseen sample by retrieving the most similar exemplar.
* Section Training of DNC: DNC directly optimizes the representation by **adjusting the arrangement** between sub-centroids and data samples.
* Section Versatility: **Reforms the training regime**
* Section Ad-hoc Explainability: explanations are **loyal to the internal decision mode**
* **stronger adhoc explainibility** - I think this sentence should be restructured.



**Typos**

This did not affect my rating,

* proximity of test data and the class sub-centroids in the feature space -> proximity of test data to the class sub-centroids in the feature space
* Parametrization nature -> parametric nature
* ImageNet [9] trained image classifiers initialize segmentation networks -> ImageNet-trained classifiers to initialize
* memory for online estimating the sub-centroids -> for estimating the sub-centroids online
* investigate post-hoc explains -> investigate classifier posthocs.
* even coming with better performance -> even with better performance
* The final output dimensionality is constrained as many as the classes. Constrained as -> constrained to be the number
* if using multiple sub-centroids (local means) per class instead of just on -> if multiple sub-centroids per class are used.
* coming with minimal architecture change -> with
* one single class weight/center is far enough -> Did you mean not enough, in the context of the results

[1] Unsupervised Learning of Visual Features by Contrasting Cluster Assignments, NeuRIPS 2020
[2] Self-labelling via simultaneous clustering and representation learning, ICLR 2020

**Strength And Weaknesses:**

**Strengths**

* The paper shows strong results on both classification and segmentation benchmarks. Ablations are shown to evaluate the impact of #subcentroids, showing that this design choice is important, i.e with no sub-centroid clustering, the algorithm performs worse than the classification layer.
* While a number of architectural changes have been proposed to neural networks, this paper proposes to modify just the last layer with performance improvements, which is of interest to the ICLR audience.
* The model provides out-of-the-box interpretability under some constraints. A few epochs before convergence, the authors propose to anchor each sub-centroid to the closest training image which can provide explainable predictions.

**Weaknesses**

See below for detailed feedback.

**Summary Of The Paper:**

The paper proposes deep nearest centroids, a new non-parametric classification layer. For each class, a number of sub-centroids are learned during the course of training. At test-time, the prediction is the class corresponding to the closest sub-centroid. The sub-centroids are learned in an iterative fashion: 1) Network parameters are updated to minimize the distance between each sample and the closest sub centroid of the ground truth class 2)  The subcentroids are updated using “momentum updates” using the centroid assignments of the current minibatch + a memory bank.

Improvements are shown on CIFAR-10 (ResNet backbones) and ImageNet (ResNet + Swin backbones). Results on semantic segmentation are even stronger on 3 algo + backbone combinations. If the sub-centroids are restricted to training images, the algorithm shows out-of-the-box interpretability, in the sense that these sub-centroids can be leveraged to explain classification decisions by the model.


**Summary Of The Review:**

I initally rate this paper below the border, even though the results are strong. See above for a list of my concrens.
I am happy to adjust my rating if the authors can address my concerns with respect to the novelty and clarity of the paper and update the draft as a result.

---

> ### Author Response · Authors · 2022-11-18
> **Point-to-Point Response to Reviewer k9FP (4/4)**
>
> **Typos:** Our apologies. They are now corrected.
>
> 1. "proximity of test data and the class sub-centroids in the feature space -> proximity of test data to the class sub-centroids in the feature space"
> 2. "Parametrization nature -> parametric nature"
> 3. "ImageNet trained image classifiers initialize segmentation networks -> ImageNet-trained classifiers to initialize"
> 4. "memory for online estimating the sub-centroids -> for estimating online the sub-centroids"
> 5. "investigate post-hoc explains -> investigate post-hoc explanations"
> 6. "even coming with better performance -> even with better performance"
> 7. "constrained as -> constrained to be the number of classes"
> 8. "if using multiple sub-centroids -> if multiple sub-centroids per class are used"
> 9. "coming with minimal architecture change -> with minimal architecture change"
> 10. "one single class weight/center is far enough -> one single class weight/center is far from enough"
>
> We appreciate again your thoughtful review and we hope we addressed your concerns. Please let us know if you'd like any further information.

---

> > ### Author Response · Authors · 2022-11-22
> > **Review acknowledgment and open dialogue**
> >
> > Your insightful comments are greatly appreciated. We have provided point-by-point responses to your concerns and are eager to engage in an open dialogue regarding them. Looking forward to hearing from you.
> >
> > Thanks.

---

> > > ### Author Response · Authors · 2022-12-01
> > > **Looking forward to the discussion**
> > >
> > > Dear reviewer,
> > >
> > > We have provided point-to-point responses to your comments. However, we have not received any feedback yet since the open discussion phase. We appreciate your new inputs based on our responses.
> > >
> > > Thank you!

---

> ### Author Response · Authors · 2022-11-18
> **Point-to-Point Response to Reviewer k9FP (3/4)**
>
> #### **Q9. Colloquail terms.**
> Thanks for your careful review. We make detailed clarification below.
>
> ---
>
> **A9.1 "Case-based":** A critical property of Nearest Centroids is its exemplar-/case-based reasoning nature [5, 11, 27]. DNC inherits this property -- ''it merely relies on the proximity of test query to local means of training data (''quintessential'' past observations) in the deep feature space''. When anchoring the sub-centroids to training images, this property becomes more apparent -- the classification is based on the similarity to these anchored observations (i.e., exemplar cases).
>
> ---
>
>
> **A9.2 "Detached...":**  As pointed out by [4, 43], black-box networks learn abstract parameters. Due to their strong learning ability, they can build the direct mapping from the input to the output, while less considering the physical/intrinsic nature of the problem being modelled. DNC is a more transparent classifier, as it clearly explains classification as the proximity of test data to class sub-centroids.
>
> ---
>
>
> **A9.3 "Limitations of parametric counterparts":** As we discussed in Sec. 1 and Sec. 3, DNC addresses several inherent limitations of parametric classifier:
> 1. Non-transparent classification decision-making process (see Fig. 1).
> 2. Lack of explicit modeling of the latent data structure. Only one single weight vector and bias term is learned for each class, less tolerant of intra-class variance.
> 3. The use of the softmax layer makes the loss only depend on the relative relation among logits (as discussed in **A6**) and harms the transferability (as discussed in **A3**).
>
> We revise the sentence as "it is fully aware of the aforementioned limitations of parametric counterparts".
>
> ---
>
>
> **A9.4 "Non-transparent parameters":** The softmax classifier is learned in a fully parametric manner. The parameters are non-transparent/non-interpretable.  The class representatives of DNC have clear physical meaning -- they are class centroids, or even real typical observations (Fig. 3).
>
> ---
>
> **A9.5 "Classification evidence":** As we repeatedly state in our paper, DNC conducts classification by ''retrieving the nearest sub-centroids''. The retrieved nearest sub-centroids are used as reference/evidence for classification.
>
> ---
>
> **A9.6 "Adjusting the arrangement":** As we discussed in **A6**, DNC directly optimizes similarities between data samples and class centers on the representation space (cf. Eq. 7), i.e., "DNC directly optimizes the representation by adjusting the arrangement between sub-centroids and data samples".
>
> ---
>
> **A9.7 "Reforms the training regime":** DNC makes a hybrid training strategy: simultaneously conducting  i) class-wise clustering (Eq. 5) for automatic sub-centroid (sub-class pattern) discovery, and ii) sub-centroid based classification for supervised representation learning (Eq. 7). This is greatly different from current training strategy of parametric classifiers.
>
> ---
>
> **A9.8 "Loyal to the internal decision mode":** DNC provides ad-hoc explainability -- DNC itself is interpretable. We can precisely know what actually makes DNC arrive at its decisions (see Sec. 4.3, Fig. 3, Fig. 4). For example, DNC indeed uses the sub-centroid images presented in Fig. 3 as reference for classification. In contrast, most network interpretation techniques only produce posteriori explanations for already-trained DNNs, typically by analysis of reverse-engineer importance values [28–30, 60–65] and sensitivities of inputs [66–69]. As many literature outlined, post-hoc explanations are problematic and misleading [32, 43, 70, 71]; they CANNOT explain what actually makes networks arrive at their decisions [72]. Essentially, post-hoc interpretation techniques explain a black-box using local approximations; they cannot truly reflect the inner working mode of  black-box networks. [43] provides extensive discussion regarding this point (see "Principle 5" in page 9-10).
>
> ---
>
> **A9.9 "Stronger ad-hoc explainability":** Thank you for pointing this out. We correct this as "... one may expect that DNC will gain enhanced ad-hoc explainability".

---

> ### Author Response · Authors · 2022-11-18
> **Point-to-Point Response to Reviewer k9FP (2/4)**
>
> #### **Q6.  "cannot directly supervise on the representation ".**
>
> **A6:** Please bear with our clarification below. This issue has been extensively discussed in many metric learning literature such as [82, 83]. As a direct quotation from [82], "... the softmax function ... cannot directly supervise on the learned representations." When considering Eq. 2, the loss does not optimize the feature representation directly, it only depends on the RELATIVE relation among class logits. When the distance between a sample and its corresponding class is relatively smaller than its distances to other classes, the penalty from the softmax loss could be marginal, but the real distance to its corresponding class might still be large. For example, for a training sample $x$ and its corresponding class $c$, its representation **x** may be far from the corresponding class weight and has a very small class score, e.g., $s^c=0.00001$, for class $c$. However, when **x** is even further from other class weights, possibly having super small class score $s^{c'}=0.00000001$ for class $c', the loss in Eq. 2 is very small –- failing to push the representation **x** close to its corresponding class. This is also one of the primary motivations of integrating metric learning into softmax classifier [82, 83]. For our case, DNC itself is a distance-/metric-based classifier -- it drops the learnable softmax layer and directly compares data samples and class representatives on the feature space (see Eq. 7). Thus it can directly optimize the feature space.
>
> ---
>
> #### **Q7. External memory.**
>
> **A7:** During the training of DNC-ResNet50 on ImageNet, for a memory with \#batch=1000, it stores 8\#gpu $\times$ 32\#batch size $\times$ 1000\#batch = 256000 \#examples.
>
> Below we give GPU memory cost (per GPU usage) with the number of class centroids $K$ and the external memory size, gathered during the training of DNC-ResNet50 on ImageNet, using eight V100 GPUs.
> | $K$ (Fixed memory size: 1000 batches (=256000 image samples))|  GPU memory cost (GB per GPU) |
> | :-: | :-: |
> | 1 | 16.07 |
> | 2 | 19.04 |
> | 3 | 22.03 |
> | 4 | 26.31 |
>
> | memory size (Fixed $K=4$) |  GPU memory cost (GB per GPU) |
> | :-: | :-: |
> | 400 batches (102400 image samples)| 11.88 |
> | 600 batches (153600 image samples)| 16.35 |
> | 800 batches (204800 image samples)| 21.65 |
> | 1000 batches (256000 image samples)| 26.31 |
>
> In our experiment, we set the size of the external memory as 256000 image samples (i.e., 1000 batches) and $K=4$ for DNC-ResNet50. Related discussion and statistics are added in Appendix Sec. K and Table 24. Thanks.
>
> ---
>
> #### **Q8. W and b.**
>
> **A8:** Thanks for pointing this out. This claim is too strong. This sentence has been revised as "W and b are learned as flexible parameters, lacking explicit modeling of the underlying data structure."

---

> ### Author Response · Authors · 2022-11-18
> **Point-to-Point Response to Reviewer k9FP (1/4)**
>
> We thank reviewer k9FP for the valuable time and constructive feedback. We provide point-to-point response below.
>
> ---
>
> #### **Q1. ... cluster assignments done in unsupervised learning.**
>
> **A1:** Thanks for your reminder. Yes, some of our techniques are indeed relevant to these clustering based unsupervised learning algorithms [ref1, ref2]. But there are also many differences, as the reviewer summarized. To address your concern, we additionally cite many recent clustering based unsupervised learning algorithms [94–101]. In page 5, we also add discussion regarding this point "This automatic sub-class discovery process also enjoys a similar spirit of recent clustering based unsupervised representation learning [24, 94–101]. However, it works in a class-wise manner, since the class label is given." Due to the page limit, we provide literature review and more detailed discussion in Appendix Sec. N.
>
> [ref1] Unsupervised Learning of Visual Features by Contrasting Cluster Assignments, NeuRIPS 2020
>
> [ref2] Self-labelling via simultaneous clustering and representation learning, ICLR 2020
>
> ---
>
> #### **Q2. Reproducibility (1. pseudocode; and 2. hyperparameters)**
>
> **A2:** Thanks for the suggestion. We would like to clarify that we had already provided pseudocode (cf. Appendix Sec. A and Alg. 1) in Appendix, which outlines each of the suggested components (e.g., the cluster assignments, parameters, memory bank, and cluster updates).
>
> Regarding $K$ and $\mu$, we just follow the common practice in the fields of image classification and segmentation to verify their values.  For instance, well-known image classifiers [3, 25, 49, 93] conduct both hyperparameter study and performance comparison on ImageNet val. In the same vein, concurrent segmentation models [3, 7, 34-35] also conduct ablation studies and performance comparisons on ADE20K val. We are aware that this protocol may not be perfectly rigorous, but we intend to adhere to it in order to ensure fairness and make the results comparable.
>
> ---
>
> #### **Q3. Transferability.**
>
> **A3:** Sorry for this confusion. For softmax classifier, both the feature network and softmax layer are fully learnable parameters. That means, for a new task, the softmax classifier has to both finetune the feature network and train a new softmax layer. However, for DNC, the feature network is the ONLY learnable part. The class centers are not freely learnable parameters; they are directly computed from training data on the feature space. For a new task, DNC just needs to fine-tune the only learnable part --  the feature network; the class centers are still directly computed from the training data according to the clustering assignments, without end-to-end training. Hence DNC owns better transferability during network fine-tuning. We provide more discussion regarding this point in Appendix Sec. F.
>
> ---
>
> #### **Q4. Sinkhorn $vs$ K-means.**
>
> **A4:** Good comment! Sinkhorn-Knopp clustering is a crucial part of our algorithm. Below we show the Top-1 score and training time when using K-means and Sinkhorn-Knopp, based on ResNet50 backbone architecture. We can find that DNC with Sinkhorn performs much better and is much more training-efficient. The experimental results are added in Appendix Sec. H and Table 17. Thanks.
>
> | Datasets  | **DNC**-K-means  | **DNC**-Sinkhorn |
> | :-: | :-: | :-: |
> | CIFAR-10 |  93.88%  | **95.78**%  |
> | CIFAR-100 | 77.86% | **79.91**%  |
>
> | Datasets  | **DNC**-K-means [Training time] | **DNC**-Sinkhorn [Training time] |
> | :-: | :-: | :-: |
> | CIFAR-10 |  4.5 hours | **2.5** hours  |
> | CIFAR-100 | 16.3 hours | **3.7** hours |
>
> ---
>
> #### **Q5. Anchor of sub-centroids.**
>
> **A5:** Thanks for your careful review. We find the closest image over the entire training set. Only considering the memory bank is less favored as the memory only stores a portion of the training samples -- we observe slight performance drop in such case.

---

### Official Review · Reviewer_GtUv · 2022-10-20

**Confidence:** 4
**Correctness:** 3
**Technical Novelty And Significance:** 2
**Empirical Novelty And Significance:** 3
**Recommendation:** 6

**Clarity, Quality, Novelty And Reproducibility:**

# Clarity
The paper is clearly written and easy to follow, but the quality of writing can surely be improved. There are numerous typos, sentences that are unclear or poorly formed.
Examples include
- page 1: "greatly boots pixel recognition"
- page 1: "... and ever popularity [20-23], Nearest Centroids, and particularly..."
- page 4: "... cannot providing..."
- Eq 5: the clustering objective that is stated suggests using dot-product as a score to determine cluster assignments and probably L2 normalised feature vectors. This is not explicitly stated, however, at this point in the paper.

# Quality
Some claims in the paper seem too strong, and the contributions are not clearly identified.
For example, the introduction claims that
-linear classifiers assume unimodality of the data "...bearing no intra-class variation.", which is not substantiated.
- "linear classifiers are trained purely for classification accuracy":  this also holds for centroid based classifiers, their difference lies in the pararmeterization of the classifier.
- On page 2 there is a list of 5 attractive properties of centroid-based classification: these seem generic and not associate with the contributions made by the authors in the current paper. The list of experimental findings in the last paragraph of page 2 (running over to page 3) seem to be the claimed contributions of the paper.

# Novelty
The paper evaluates nearest centroid classifiers in combination with deep feature learning networks.
The nearest centroid classifier, as well as the optimal-transport based clustering approach are known in the literature. Their combination, however, appears novel to me.
I haven't seen nearest centroid classifiers for semantic segmentation, but I'm not actively working on this  application nor on centroid-based classifiers, so I could have missed something.
In particular the transfer ability of the imageNet pertained feature backbone to the segmentation task is an interesting experiment in the paper, that contributes to the literature.

# Reproducibility
The work seems fairly easy to reproduce. The method is clearly described. Code is promised, but not provided. Experiments were conducted across an 8 V100 GPU hardware setup.
It wasn't clear to me in Section 4.4 why using K>4 centroids per class was not possible for the imageNet classification experiments, but it was possible to go up to K=20 for the ADE20K semantic segmentation experiment. It would be useful to add a paragraph analysing the memory cost of the model wrt the number of centroids K and the "external memory" size used to store additional batches for clustering.


**Strength And Weaknesses:**


Strengths:

+ Two different tasks are considered to evaluate the effectiveness of deep nearest centroids classifiers: image classification and semantic segmentation (pixel classification).

+ For each task two datasets are considered: ImageNet, CIFAR-10 for classification, and ADE20K, Cityscapes for semantic segmentation.

+ Multiple network architectures are considered for each experiment.

+ Exploration of the use of training samples rather than cluster centres as anchors for classification is interesting from an interpretability perspective.

+ The use of nearest centroid classification is shown the be beneficial for semantic segmentation: (i) using it to fine-tune the segmentation model, and (ii) using it to pre-train the feature backbone network. Using it for both yields best performance.


Weaknesses:

- The claimed improved transferability is shown when pre-training on ImageNet classification, and then fine-tuning on ADE20K/Cityscapes semantic segmentation. There is no experimental evidence that transferability is improved towards other image classification tasks. Such results would be relatively easy to add, and would strengthen the support of the claim of improved transfer of features learned using the centroid-based classifier.

- Experimental results do not include analysis of the stability of the results w.r.t. various sources of randomness in the experimental setup (initialisation of parameters, sampling of training batches). In particular for tables 1 and 2 this would help, as differences in accuracy are relatively small.  For ImageNet classification adding evaluation on the ImageNetv2 test set (https://github.com/modestyachts/ImageNetV2) would also strengthen the experimental results.

- It is not clear where the normalised exponential for of the solution presented in Eq 6 comes from. In my understanding it is a consequence of the KL-regularization used in [25], but which is not adopted here in Eq 5. I guess it is an omission.

- The effect of the optimal-transport / Sinkhorn clustering approach as compared to a naive k-means approach is not ablated in the paper, while this is the main novelty of the paper over [52] besides using multiple centroids per class.

- The setting of the temperature parameter epsilon in Eq 6 is not discussed, nor is its impact experimentally analysed.

- Relations of the current work to [23,52] are not discussed in sufficient detail, it should be clear from the paper how this work is different from these earlier works. For example, the related work section states that [52] showed weak results even on small datasets, yet it is not clear why the current paper would work better for those cases: the ablations show only limited improvements over the case with K=1 cluster center per class as used in [52]. It is not clear from this why for the current paper experimental comparison to [52] is not included.



**Summary Of The Paper:**

This paper considers the use of nearest centroid classifiers in the context of deep neural networks, and in particular a setting where multiple centroids are used per class, obtained through clustering the feature vectors for each class.
The proposed method consists in concurrently learning the feature backbone network, and a per-class online clustering. For the clustering an optimal-transport based method is used to approximately equalise the amount of data assigned to each cluster.
Experiments on image classification (ImageNet, CIFAR-10) and semantic segmentation (ADE20K, Cityscapes) are conducted using different network architectures.
Results indicate improvements in prediction accuracy when comparing parametric linear classifier heads and nearest centroid classification heads. Moreover, networks pre-trained with the nearest centroid classifier, also transfer better to semantic segmentation tasks.
In addition, when using data points close to the cluster centres for classification, the method allows for interpretable results via visualisation of the exemplar used for classification.


**Summary Of The Review:**


This paper considers nearest centroid classifiers in combination with deep feature learning networks. The authors train the feature network, and cluster the data of each class in parallel using an optimal-transport based technique: a novel combination of existing techniques.
Experiments on image classification and semantic segmentation compare the centroid-based classifier with a linear classification head, and conclude improved accuracy using centroid-based classifiers.
Experiments also indicate that centroid-based pre-training for image classification yields better results when fine-tuning for semantic segmentation.
Replacing centroids with the nearest datapoints (images) induces a small loss in predictive accuracy, but yields an interpretable exemplar-based classifier.
My ratings are based on a small perceived technical novelty, combined with some interesting experimental findings. The quality of the presentation can be substantially improved.

### Post-rebuttal comments ###
After reading author rebuttal and other reviews, I maintain my original recommendation of this paper as weak accept.
The authors have addressed most of the comments in my review: a number of clarifications have been made, and additional results are added, in particular evaluation on the ImageNetv2 evaluation sets, which confirm the results obtained earlier on the imageNet validation set.  I do not see any major reasons to reject the paper.

---

> ### Author Response · Authors · 2022-11-18
> **Point-to-Point Response to Reviewer GtUv (3/3)**
>
> **Quality:** Our apologies. We work to improve our writing.
>
> 1. unimodality assumption: Parametric softmax learns one single weight (and a bias term) per class. That means it relies on an implicit assumption of unimodality of data of each class in the feature space. However, this unimodality assumption is rarely the case in real-world scenarios and makes the model less tolerant of intra-class variances. This issue has been pointed out by [6, 7, 23, ref6, ref7].
> 2. "...bearing no intra-class variation" -> "...less tolerant of intra-class variation".
> 3. "linear classifiers are trained purely for classification accuracy". In addition to optimizing classification accuracy, DNC needs to automatically discover sub-class patterns. Thus, in our case, the one-hot assignment constraint and equipartition constraint in the clustering target (Eq. 5) complement the classification accuracy based training objective. To avoid potential misleading, this sentence is revised as "linear classifiers are typically trained to optimize classification accuracy only, paying less attention to modeling the latent data structure".
> 4. "... 5 attractive properties". We do not claim these properties are our unique characteristics or contribution. These statements are just to help readers to better understand our advantages over parametric counterparts. Moreover, as far as we know, there is no previous centroid-based classifier that can simultaneously show these properties. For instance, most prototype-/centroid-based classifiers [4, 5, 72–74, 79, 80] adopt softmax classifier's pre-trained weight. That means they have to first build and train a softmax classifier, based on which then they can deliver a prototype-based classifier. Hence their interpretability is post-hoc, and their transferability is rather weak or quite unclear. Differently, DNC is trained on ImageNet from scratch and clearly shows its strong transferability. In addition, many prototype-based classifiers bring significant architectural change and some of them, like [72, 74, 79], are even task-specific. It is thus hard to say simplicity and transparency are generic for prototype-/centroid-based models.
>
> [ref6]  Feedforward neural networks initialization based on discriminant learning. Neural Networks 2022
>
> [ref7]  Neural Collapse Under MSE Loss: Proximity to and Dynamics on the Central Path. ICLR 2021
>
> ---
>
> **Typos:** Thank you so much for your careful review! They are fixed and a thorough proofreading is made.
>
> 1. "greatly boots pixel recognition -> greatly boosts pixel recognition"
> 2. "Nearest Centroids, and particularly ... -> Nearest Centroids and its utility in large datasets with high-dimensional input spaces are ..."
> 3. "cannot providing -> which cannot provide"
> 4. Eq 5: Yes, L2 normalization is used. We add a statement that "For simplicity, all the features are defaulted to l2-normalized from now on" under Eq. 3.
>
> ---
>
> **Novelty:** "I haven't seen nearest centroid classifiers for semantic segmentation."
>
> Yes, to our best knowledge, we take the lead to apply Nearest Centroid classifier for segmentation. Our DNC boosts segmentation performance by +**1.1-2.5**% mIoU. This is a quite big score for the segmentation task. For example, on Cityscapes, a recent art [ref6] only improves DeepLabV3 by +1.1%, while DNC gives +**1.7**%. We even view this as a small piece of our contribution and do not address it in our paper. Thanks again for your careful review.
>
>
> [ref6] Exploring Cross-Image Pixel Contrast for Semantic Segmentation, ICCV 2021 Oral
>
> ---
>
> Finally, we would like to sincerely thank you for your careful review and valuable comments, which really help us to improve this work. We hope we addressed your concerns. Please let us know if you'd like any further information.

---

> > ### Author Response · Authors · 2022-11-22
> > **Review acknowledgment and open dialogue**
> >
> > Your insightful comments are greatly appreciated. We have provided point-by-point responses to your concerns and are eager to engage in an open dialogue regarding them. Looking forward to hearing from you.
> >
> > Thanks.

---

> > > ### Author Response · Authors · 2022-12-01
> > > **Looking forward to the discussion**
> > >
> > > Dear reviewer,
> > >
> > > We have provided point-to-point responses to your comments. However, we have not received any feedback yet since the open discussion phase. We appreciate your new inputs based on our responses.
> > >
> > > Thank you!

---

> ### Author Response · Authors · 2022-11-18
> **Point-to-Point Response to Reviewer GtUv (2/3)**
>
> #### **Q6. Relations to current work.**
>
> **A6:** Thanks for your careful review. [22] is NOT a deep learning approach. As for [52], it also investigates the idea of bringing Nearest Centroids into DNNs. However, there are several novel differences between [52] and DNC, which cause significant performance gap: First, [52] simply abstracts each class into one single class mean, failing to capture complex
> within-class data distribution. In contrast, we consider $K$ sub-centroids per class. Note that this is not just increasing the number of class representatives. This requires accurately discovering the underlying data structure. We therefore jointly conduct automated online clustering (for mining sub-class patterns) and supervised representation learning (for cluster center based classification). Finding meaningful class representatives is extremely challenging and crucial for Nearest Centroids. As the experiment in **A4** shown, simply adopting classic $k$-means causes huge performance drop and significant training speed delay.  Second, [52] computes the class mean on each batch, which makes a poor approximation of the real class center. In contrast, we adopt the external memory for more accurate data density modeling -- our clustering is made over the large memory, instead of the relatively small batch. Therefore, the baseline $K=1$ in our ablation (Table 5a) DOES NOT correspond to [52].  Third, to better address your concern, we compare DNC with [52] on CIFAR-10 and CIFAR-100 (as [52] only describes its training procedures for CIFAR-10 and CIFAR-100). As seen, DNC greatly outperforms [52]: +**2.11**% on CIFAR-10 and +**7.16**% on CIFAR-100. The experimental results and related discussions are added in Appendix Sec. I and Table 18.
>
> | Datasets  | [52]  | **DNC**-ResNet50 |
> | :-: | :-: | :-: |
> | CIFAR-10 |  93.67%  | **95.78**%  |
> | CIFAR-100 | 72.75% | **79.91**%  |
>
>
> As for "It is not clear from this why for the current paper experimental comparison to [52] is not included", it is just because we thought the performance of [52] is rather weak, and hence we put our whole focus on the comparison to the strong parametric softmax competitor. But we should indeed report this experiment. Thank you again for your careful review!
>
> ---
>
> #### **Q7. Memory cost.**
>
> **A7:** Sorry for this confusion. As segmentation is pixel classification, each training image provides numerous pixel samples for each class. For image classification, however, each training image is only assigned to one single class. Moreover, ImageNet has 1000 classes, while in general semantic segmentation only has dozens of classes. Therefore, for a training mini-batch of, for example, 256 images, every class in segmentation usually has many training **pixel** samples in each mini-batch; this allows us to use a large K for clustering. However, under the same setting, for each mini-batch, there must have many ImageNet classes that do not have corresponding **image** samples -- we have 1000 classes but each mini-batch only has 256 training images. This is why we need to build an external memory during ImageNet classification. Even with external memory, our current machine can only afford $K=4$  (we use eight V100 GPUs, which are the best machine in our lab). This is also why applying Nearest Centroids for batch-wise ImageNet classification training is extremely challenging; [52] cannot handle ImageNet classification, as it only computes class means in a batch-wise manner.
>
> Below we give GPU memory cost (per GPU usage) with the number of class centroids $K$ and the external memory size, gathered during the training of DNC-ResNet50 on ImageNet, using eight V100 GPUs.
> | $K$ (Fixed memory size: 1000 batches (=256000 image samples))|  GPU memory cost (GB per GPU) |
> | :-: | :-: |
> | 1 | 16.07 |
> | 2 | 19.04 |
> | 3 | 22.03 |
> | 4 | 26.31 |
>
> | memory size (Fixed $K=4$) |  GPU memory cost (GB per GPU) |
> | :-: | :-: |
> | &#8194;400  batches (102400 image samples)| 11.88 |
> | &#8194;600  batches (153600 image samples)| 16.35 |
> | &#8194;800  batches (204800 image samples)| 21.65 |
> | 1000 batches (256000 image samples)| 26.31 |
>
> In our experiment, we set the size of the external memory as 256000 image samples (i.e., 1000 batches) and $K=4$ for DNC-ResNet50 (i.e., a total of 4000 class sub-centroids for ImageNet). In comparison, for segmentation, when we set $K=10$ (i.e., a total of 1500 class sub-centroids for ADE20K), we have 65536 pixel training samples (on the feature level) in each mini-batch of 16 training images, without using memory. Related discussion and statistics are added in Appendix Sec. K and Table 24.

---

> ### Author Response · Authors · 2022-11-18
> **Point-to-Point Response to Reviewer GtUv (1/3)**
>
> We thank reviewer GtUv for the valuable time and constructive feedback. We provide point-to-point response below.
>
> ---
>
> #### **Q1. Transferability towards other image classification task.**
>
> **A1:** Great suggestion! First, we had already provided such kind of experiment, i.e., exploring transferability on image classification task, in Appendix. In Sec. E, we follow the experimental setup posed by [106] –– learning the feature embedding using coarse-grained object labels, and evaluating the learned feature using fine-grained object labels. From Table 13 and 14, we can find that DNC clearly performs better than the parametric classifier in such coarse-to-fine transfer learning experiment.
>
> Second, we agree that adding more experiment on other classification datasets can better support our claim. We further evaluate the transferability by applying ImageNet-trained weight to Caltech-UCSD Birds-200-2011 (CUB-200-2011) dataset [ref1], as in [ref2-ref4]. Below results show that, compared with the parametric counterpart, DNC is +**0.73**% and +**0.39**% higher in Top-1 and Top-5 acc., respectively. This experiment is added in Appendix Sec. F and Table 15. Thanks.
>
> | Dataset  | ResNet50 (Top-1, Top-5)  | **DNC**-ResNet50 (Top-1, Top-5)|
> | :-: | :-: | :-: |
> | CUB-200-2011 |  84.48%, 96.31%  | **85.21**%, **96.70**%  |
>
> [ref1] The Caltech-UCSD Birds-200-2011 Dataset, 2011
>
> [ref2] Self-tuning for data-efficient deep learning, ICML 2021
>
> [ref3] Deep Metric Learning via Lifted Structured Feature Embedding, CVPR 2016
>
> [ref4] Deep transfer learning for image classification: a survey, arXiv 2022
>
> ---
>
> #### **Q2.1 Stability.**
>
> **A2.1:** Sorry for this confusion.  We clarify that we had already provided error bars in Table10&11&12 in the Appendix Sec. D (based on 3 runs). The scores correspond to our classification and segmentation results in Table1&2&3. We can safely conclude that the improvements brought by DNC are solid and consistent. Thanks.
>
> ---
>
> #### **Q2.2 Evaluation on the ImageNetv2 test set.**
>
> **A2.2:** Good comment! Following your suggestion, we evaluate ResNet50-DNC on the ImageNetv2 test sets [ref5], i.e., "Matched Frequency", "Threshold0.7" and "Top Images". As seen, DNC consistently exceeds the parametric softmax ResNet50 by +**0.52-0.89**% top-1 and +**0.26-0.47**% top-5 acc. across all the test sets. This experiment is added in Appendix Sec. G and Table 16. Thanks.
>
> | ImageNetv2 testsets  | ResNet50 (Top-1, Top-5)  | **DNC**-ResNet50 (Top-1, Top-5)  |
> | :-: | :-: | :-: |
> | MatchedFrequency |  63.30%, 84.70%  | **63.96**%, **85.17**%  |
> | Threshold0.7 | 72.70%, 92.00% | **73.59**%, **92.26**%  |
> | TopImages | 78.10%, 94.70% | **78.62**%, **94.96**%  |
>
> [ref5] Do ImageNet Classifiers Generalize to ImageNet?, ICML 2019
>
> ---
>
> #### **Q3. Normalised exponential term in Eq.6.**
>
> **A3:** Yes, Eq. 5 is relaxed to be an Optimal Transport with entropic constraints [23, 24]. Due to page limit, we omit the detail of reformulating the constraints in the entropic form. Thanks.
>
> ---
>
> #### **Q4. Sinkhorn $vs$ naive K-means**
>
> **A4:** Good comment! Using K-means is our start point. We should indeed provide such baseline. We show Top-1 score and training time, based on ResNet50 backbone architecture. We can find that DNC with Sinkhorn performs much better and is much more training-efficient. The experimental results are added into the Appendix Sec. H and Table 17. Thanks.
>
> | Datasets  | **DNC**-K-means  | **DNC**-Sinkhorn |
> | :-: | :-: | :-: |
> | CIFAR-10 |  93.88%  | **95.78**%  |
> | CIFAR-100 | 77.86% | **79.91**%  |
>
> | Datasets  | **DNC**-K-means [Training time] | **DNC**-Sinkhorn [Training time] |
> | :-: | :-: | :-: |
> | CIFAR-10 |  4.5 hours | **2.5** hours  |
> | CIFAR-100 | 16.3 hours | **3.7** hours |
>
> ---
>
> #### **Q5. Temperature parameter $\epsilon$ in Eq.6.**
>
> **A5:** $\epsilon$ in Eq.6 trades off convergence speed with closeness to the original transport problem [23, 24]. We do not extensively fine-tune this parameter; we just follow [23, 24] to set $\epsilon$ as 0.05 for all our experiments.  Below we show experimental results on ImageNet. The experimental results are added in Appendix Sec. J and Table 22. Thanks.
>
> | $\epsilon$  |  **DNC**-ResNet50 (Top-1, Top-5)|
> | :-: | :-: |
> | 0.01 |  76.34%, 92.97%  |
> | 0.05 | **76.49**%, **93.08**% |
> | 0.1 | 76.40%, 93.02% |

---

> ### Author Response · Authors · 2022-12-07
> **Seek open dialogue**
>
> Dear Reviewer GtUv,
>
> Thank you again for your kind review and comments. We hope we have addressed all of your concerns. We wonder if there are any additional comments. We sincerely hope that we will be able to use the remaining time to engage in an open dialogue with domain experts to enhance the quality of our work.
>
> Thanks again,
> Authors.

---

### Official Review · Reviewer_3KaQ · 2022-10-23

**Confidence:** 4
**Correctness:** 4
**Technical Novelty And Significance:** 3
**Empirical Novelty And Significance:** 3
**Recommendation:** 8

**Clarity, Quality, Novelty And Reproducibility:**

The paper is well written, easy to follow. Although the nearest centroid classifier itself is not a novel idea, the successful application of it to large-scale visual recognition is non-trivial. The method is clearly described and should be reproducible.

**Strength And Weaknesses:**

Strength:
++ Nearest centroids based classifiers have been used in many classification problems and exhibit strong performance compared with its parametric counterparts. This work applies the non-parametric classifier in the basic image classification task and achieves competitive performance by utilising sub-centroids of training samples to describe class distributions.
++ Thorough experiments have been conducted to support the claims in the manuscript.

Weaknesses:
-- The number of centroids for each class K seems to be an important hyper-parameter. In the paper, K is set as a unique value for all classes; however, this may not be optimal since the intra-class variability can be very different across different classes. Is it possible to automatically determine the K value for different classes?
-- The nearest centroids classifier only considers the 1st-order statistics. Would the 2nd-order statistics of clusters contribute to the classification performance?
-- It would be interesting to compare the errors made by DNC and a parametric softmax classifier. Do they learn complementary knowledge?

**Summary Of The Paper:**

This paper proposes a deep nearest centroids (DNC) network for large-scale visual recognition. In contrast to the commonly employed parametric classifier (e.g., a parametric Softmax Classifier), the proposed DNC is claimed to be able to capture the underlying data distributions and is more interpretable. Experiments on a few benchmark visual recognition and semantic segmentation datasets demonstrate the superiority of DNC.


**Summary Of The Review:**

To summarize, the paper investigates a classific non-parametric classifier and successfully integrates it into deep learning frameworks. Competitive performance have been achieved. The work can provide useful insights into representation learning and visual recognition for the community.

---

> ### Author Response · Authors · 2022-11-18
> **Point-to-Point Response to Reviewer 3KaQ**
>
> We thank reviewer 3KaQ for the valuable time and constructive feedback. We provide point-to-point response below.
>
> ---
>
> #### **Q1. Number of centroids (and is it possible to automatically determine it?).**
>
> **A1:** Thanks for pointing out this promising direction. Previously, we had ever conducted an experiment by setting $K$ with different values based on the number of training samples of each class. Specifically, ImageNet contains between 732 and 1300 training images (#images) per class. Then, $K=1$ is assigned to the class having between 732 and 874 training samples, $K=2$ to the class having between 875 and 1016 samples, $K=3$ to the class having between 1017 and 1158 samples, and $K=4$ to the class having between 1159 and 1300 samples. We can find that we gain slightly better performance, compared with fixing $K=4$ for all the classes. In our current version, we directly set $K=4$ for different classes for the sake of similarity. The experimental results are added in Appendix Sec. J and Table 23. Thanks.
>
> | $K$ range  | ImageNet (Top-1, Top-5)  |
> | :-: | :-: |
> | unique value 4 |  76.49%, 93.08% |
> | [1, 4] |  **76.55**%, **93.10**% |
>
> Also, as you noticed, it is interesting to explore how to automatically determine the $K$ values. We found that there are some clustering techniques (like [ref1]) that can automatically determine the number of cluster centers. Maybe they can serve as the basis of our future work. However, after running their code we find their algorithms are rather complicated and time-consuming. We discuss this point at Appendix Sec. O.  Any further suggestion is very welcome.
>
> [ref1] DeepDPM: Deep Clustering With an Unknown Number of Clusters, CVPR 2022
>
> ---
>
> #### **Q2. Second-order statistics.**
>
> **A2:** Thanks for your comment. Considering second-order statistics is reasonable, but it greatly improves the computational cost. We observe huge training speed delay (about 3.7 times longer on ImageNet), due to additional computing demand for the covariance. According to our experiment made during the rebuttal phase, we even observe slight performance drop on ImageNet: 76.31\% $vs$ 76.49\%. But we think this is indeed a promising direction but more engineering effort is needed to make it solid. We highlight this point as a part of our future work in Appendix Sec. O.
>
> ---
>
> #### **Q3. Compare the errors.**
>
> **A3:** Following your suggestion, we compare some images wrongly assigned by DNC and softmax classifiers. Unfortunately, we do not observe any significant pattern. But we like this idea. We will make more exploration towards this direction. For example, we will further study the robustness against perturbation, adversarial attack, and out-of-distribution data. We address this point in Appendix Sec. O. Thanks.
>
> ---
>
> We appreciate again your thoughtful review and we hope we addressed your concerns. Please let us know if you'd like any further information.

---

> > ### Author Response · Authors · 2022-11-22
> > **Review acknowledgment and open dialogue**
> >
> > Your insightful comments are greatly appreciated. We have provided point-by-point responses to your concerns and are eager to engage in an open dialogue regarding them. Looking forward to hearing from you.
> >
> > Thanks.

---

### Official Review · Reviewer_1ipB · 2022-10-27

**Confidence:** 4
**Correctness:** 3
**Technical Novelty And Significance:** 3
**Empirical Novelty And Significance:** 1
**Recommendation:** 5

**Clarity, Quality, Novelty And Reproducibility:**

The paper is clearly written, and the method design is well-constructed with proper motivations of the several choices. Novelty is related to the formalization of the method and is, in my opinion, acceptable for ICLR, but the placement of the work within the state of the art on prototype- and distance-based learning is falling short (see statements above). Some statements, especially in the introduction, are not backed with thorough experimental results or not addressed explicitly in the remainder of the paper.

I trust that reproducibility can be ensured as long as the author would publish their code, as promised.

**Strength And Weaknesses:**

__Strenghts__
- the idea of learning deep centroids is relevant
- interpretability of the classification with prototypes or a real sample closest to the prototype
- experiments with different backbones


__Weaknesses__
- _statements about limitations of existing methods are too stretched_, e.g.
	1. lack of simplicity or explainability mentioned as weaknesses of existing approaches is not addressed clearly in the paper: the mention of lack of simplicity of existing methods makes one assume that the authors would propose a simpler method. Anyway, it is still absed on gradient descent optimization, with the addition of extra steps to ensure clustering after every update of the weights. Furthermore, the memory and algorithmic complexity of the proposed solution increased, making me wonder where the lack of simplicity is addressed.
	2. learning without considering underlying data structures (i.e. by similarity) and on fixed dimensional spaces: the authors miss the relation of this paper with a body of work on similarity-based and distance learning, e.g. contrastive learning or arcloss, is not covered. In these works, learning and optimization of parameters is done on the basis of similarity/metric learning, and subsequently decisions are taken by looking at the closest samples in the latent space. Other methods based on prototype learning are also ignored, e.g. [1][2]
- _experiments should be extended_: methods based on latent similarity metric learning should be compared to, as the principle of classification/prediction for those methods is the same of closest-prototype picking.
- _result improvements are marginal, with no statistical analysis of differences or details about replication of experiments_: the results are only very marginally above the baselines, and statistical margins are not reported. Are the results from the best runs of training or e.g. median over 5 runs, average over N runs, etc.? It is difficult to evaluate the contribution of the clutering approach.
- _explainability/interpretability study needs extension and comparative analysis_: other approaches are used to do predictions based on distance/similarity in the latent space (e.g. supervised contrastive learning, circle loss, arcloss) that are not considered in the comparative analysis, also in terms of interpretability of the classifications.

[1] nauta et al. Neural Prototype Trees for Interpretable Fine-grained Image Recognition
[2] Chen et al. This looks like that: Deep learning for interpretable image recognition

**Summary Of The Paper:**

The paper proposes a method that learns deep prototype representations, by revisiting the nearest centroids clustering in the setting of deep network training. The authors construct a framework to combine the gradient descent kind of learning of neural networks with the distance-based learning of cluster centroids. In this paper, the clustering is done class-wise, defining a number of centroids K per class.

**Summary Of The Review:**

The paper presents an interesting idea, well-constructed, and with a clear presentation of the devised methodology. The main concerns with the paper regard placement within the current literature, clarification of several statements made in the introduction and experimental analysis.

This work seems to relate with metric learning, e.g. contrastive learning, approaches but the authors do not cover this part. Indeed, equation 7 reminds very clearly the 'contrastive loss function' (with the difference that only the min distance is considered in (7)), see [3] equation 1.

The reported results are only marginally 'better' than baseline networks, but the authors do not compare with other related approaches (mentioned few above), making the evaluation of the contribution of deep prototype learning difficult to appreciate. Also in the discussio nabout explainability, methods based on classification by nearest sample (arcloss/arcface-like methods, contrastive approaches) should have been considered.

[3] https://openaccess.thecvf.com/content/CVPR2021/papers/Wang_Understanding_the_Behaviour_of_Contrastive_Loss_CVPR_2021_paper.pdf

---

> ### Author Response · Authors · 2022-11-18
> **Point-to-Point Response to Reviewer 1ipB (3/3)**
>
> #### **Q3.1 DNC results are marginal.**
>
> **A3.1:** We respectfully disagree. First, DNC provides +**0.23-0.24**% and +**0.24-0.32**% top-1 acc. improvements on CIFAR-10 and ImageNet, compared with parametric softmax classifier. This is notable: as far as we know, no previous explainable classifier can give such promising results on challenging ImageNet. For example, [ref7, ref8] only report experiments on CIFAR-10, even with -0.05% and -0.04% performance drop. Second, there is no previous explainable classifiers provided experiment on both image classification and segmentation tasks; some of them are even task-specific [72, 74, 79]. Third, DNC also boosts segmentation performance by +**1.1-2.5**% mIoU. This is a quite big score for segmentation task. For example, on Cityscapes, [ref6] only improves DeepLabV3 by +1.1%, while DNC gives +**1.7**%. Thanks.
>
>
> [ref7] Interpretable part-whole hierarchies and conceptual-semantic relationships in neural networks, CVPR 2022
>
> [ref8] Adaptive activation thresholding: Dynamic routing type behavior for interpretability in convolutional neural networks, ICCV 2019
>
> ---
>
> #### **Q3.2 Statistical analysis.**
>
> **A3.2:**  Sorry for this confusion. We clarify that we have already provided error bars in Table10&11&12 in the Appendix Sec. D (using average over 3 runs). The scores correspond to our classification and segmentation results in Table1&2&3. We can safely conclude that the improvements brought by DNC are solid and consistent. Thanks.
>
> #### **Q3.3 It is difficult to evaluate the contribution of the clustering approach.**
> **A3.3:**  In Table 5a, the baseline $K=0$ refers to the performance without clustering.
>
> To better address your concern, we train DNC based on the classic K-means. We show Top-1 score and training time, based on ResNet50 backbone network architecture. We can find that DNC with Sinkhorn performs much better and is much more training-efficient. The experimental results are added in Appendix Sec. H and Table 17. Thanks.
>
> | Datasets  | **DNC**-K-means  | **DNC**-Sinkhorn |
> | :-: | :-: | :-: |
> | CIFAR-10 |  93.88%  | **95.78**%  |
> | CIFAR-100 | 77.86% | **79.91**%  |
>
> | Datasets  | **DNC**-K-means [Training time] | **DNC**-Sinkhorn [Training time] |
> | :-: | :-: | :-: |
> | CIFAR-10 |  4.5 hours | **2.5** hours  |
> | CIFAR-100 | 16.3 hours | **3.7** hours |
>
> ---
>
> #### **Q4. Relation to other approaches, e.g. supervised contrastive learning, circle loss, arcloss.**
>
> **A4:** As we explained in **Clarification**, [ref1-ref3] are just metric learning methods. They are NOT metric based classifiers. Their trained models are still **black-box** networks. They cannot provide any interpretability. For concept/prototype based explainable networks [4, 5, 72–74, 79, 80, ref5], they are post-hoc explainable models. They are essentially built upon pre-trained softmax classifiers and bring significant architectural change. Some of them, like [72, 74, 79], are even task-specific. In sharp contrast, DNC is a **general**, **ad-hoc interpretable**, and **nonparametric class centroid based** classifier. Thanks.
>
> ---
>
> We appreciate again your thoughtful review and we hope we addressed your concerns. Please let us know if you'd like any further information.

---

> > ### Author Response · Authors · 2022-11-24
> > **Review acknowledgment and open dialogue**
> >
> > Your insightful comments are greatly appreciated. We have provided point-by-point responses to your concerns and are eager to engage in an open dialogue regarding them. Looking forward to hearing from you.
> >
> > Thanks.

---

> > > ### Author Response · Authors · 2022-12-01
> > > **Looking forward to the discussion**
> > >
> > > Dear reviewer,
> > >
> > > We have provided point-to-point responses to your comments. However, we have not received any feedback yet since the open discussion phase. We appreciate your new inputs based on our responses.
> > >
> > > Thank you!

---

> > ### Comment · Reviewer_1ipB · 2022-12-09
> > **response appreciated**
> >
> > dear,
> > I appreciate the response, further clarifications to my doubts, and extra results provided.
> >
> > For the sake of completeness, I cannot see the results for K=0 in Table 5a, which I agree with the authors that would indeed give a complete understanding of the effects on the performance determined by clustering. Am I missing something?
> >
> > I definitely appreciate the extent and depth of the response.

---

> > > ### Author Response · Authors · 2022-12-09
> > > **Thanks for your careful review**
> > >
> > > We are sorry for the typo here. $K=1$ in Table 5a should be the results without clustering. Direct quote from our paper "When K = 1, each class is represented by its centroid – the average feature vector of all the training samples of the class (Eq. 3), **without clustering**."
> > >
> > > Thanks again for your careful review. Please let us know if any other concerns.

---

> ### Author Response · Authors · 2022-11-18
> **Point-to-Point Response to Reviewer 1ipB (2/3)**
>
> #### **Q2. Experiments on metric learning.**
>
> **A2:** To address your concern, we compare DNC with [52] using ResNet50 backbone architecture.  [52] conducts similarity-based classification using class means. As [52] only describes its training procedures for CIFAR-10 and CIFAR-100, we make the comparison on these two datasets to ensure fairness. Below results show that, DNC significantly outperforms [52] by +**2.11**% on CIFAR-10 and +**7.16**% on CIFAR-100, respectively. [52] only makes use of one single center per class and estimates the class center per batch. In contrast, DNC explores within-class clustering on the external memory for automatically discovering $K$ representative class sub-centroids. Thus DNC better captures complex within-class variants and addresses two key properties of prototypical exemplars: sparsity and expressivity [58, 59].
>
> | Dataset  | [52]  | **DNC**-ResNet50 |
> | :-: | :-: | :-: |
> | CIFAR-10 |  93.67%  | **95.78**%  |
> | CIFAR-100 | 72.75% | **79.91**%  |
>
> We also compare DNC with [49] using ResNet50 backbone architecture. [49] is a deep $k$-NN classifier, which conducts similarity-based classification based on top-$k$ nearest training samples. As [49] only reports results on ImageNet, we make the comparison on ImageNet to ensure fairness. Note that 130 training epochs are used, as in [49]. Below results show that, DNC outperforms [49]. Note that [49] poses huge storage demand, i.e., retaining the whole ImageNet training set (i.e., 1.2M images) to perform the k-NN decision rule, and suffers from very low efficiency, caused by extensive comparisons between each test sample and ALL the training images.  These limitations prevent the adoption of [49] in real application scenarios. In contrast, DNC only relies on a small set of class representatives (i.e., four sub-centroids per class) for decision-making and causes no extra computation budget during network deployment.
>
> | Dataset  | [49]  | **DNC**-ResNet50 |
> | :-: | :-: | :-: |
> | ImageNet |  76.57%  | **76.64**%  |
>
> We further compare our DNC with [ref6] on Cityscapes val set, using ResNet101 backbone and DeepLabV3 segmentation architecture. [ref6] applies contrastive learning to segmentation. Please note that [ref6] still relies on softmax classifier. Thus it is in essence a metric learning boosted pixel-wise softmax classifier; it is NOT a distance-based classifier. Below results show that, DNC greatly surpasses [ref6] by 0.6% mIoU.
>
> | Datasets  | [ref6]  | **DNC**-DeepLabV3 |
> | :-: | :-: | :-: |
> | Cityscapes |  79.2%  | **79.8**%  |
>
> These experiments solidly demonstrate our effectiveness on both image classification and segmentation tasks, even compared with other distance (learning) based counterparts. Related experiments are added in Appendix Sec. I, Table 18&19&20. Thanks.
>
> [ref6] Exploring Cross-Image Pixel Contrast for Semantic Segmentation, ICCV 2021

---

> ### Author Response · Authors · 2022-11-18
> **Point-to-Point Response to Reviewer 1ipB (1/3)**
>
> We thank reviewer 1ipB for the valuable time and constructive feedback. We provide point-to-point response below.
>
> ---
>
> #### **Q1.1 Simplicity/explainability**
>
> **A1.1:** Sorry for this confusion. We have clearly mentioned that DNC is a neural network classifier. It seems quite natural to understand that DNC adopts gradient descent optimization.
>
> We use the term "simplicity" to refer to DNC's intuitive classification mode. It strictly follows the easy-to-understand decision mechanism of Nearest Centroids: a test sample is directly classified into the class with closest centroids. There is broad agreement that Nearest Centroids is one of the simplest classifiers [10–15]. In addition, from the perspective of network architecture, DNC is elegant, as it drops the softmax layer. To learn such a beautiful classifier, we present an appealing hybrid of online unsupervised sub-class pattern discovery and end-to-end supervised classification training. This well addresses the nature of Nearest Centroids and brings novel insights into the visual recognition task itself. In addition, external memory and clustering are no longer needed after training. They only cause 5\% delay during ImageNet training, yet no any extra computation budget during inference.
>
> Please note that the interpretability of a classifier mainly comes from its decision-making mode, instead of the training strategy. We have provided extensive experiments (Sec. 4.3), many visual examples (Fig. 3 and 4), and interpretable IF ... Then rules (Eq. 9 and 10) to examine our interpretability. Such interpretability is particularly valuable considering that DNC even shows better performance on large-scale and challenging ImageNet classification and semantic segmentation tasks. Thanks.
>
> ---
>
> #### **Q1.2 Relation to similarity-based and distance learning approaches:**
>
> **A1.2:** We would like to clarify we had already discussed our relations to numerous similarity-/distance-based classifiers [49, 51, 52, 53, 54, 55, 56, 57] and concept-/prototype-based explainable networks [4, 5, 72–74, 79, 80] in the first two subparagraphs of Related Work ([ref5] had been cited as [72]). As distance/similarity learning is a quite huge field, we have to put our focus on the most relevant ones.
>
>
> But we are happy to make further detailed discussion to better address your concern. DNC is fundamentally different from [ref4, ref5]: i) [ref4, ref5] need to **LEARN** prototypes in a fully parametric manner; they are post-hoc explainable models. DNC instead directly sets the prototypes as class sub-centroids and can anchor prototypes to real observations. DNC is ad-hoc interpretable. ii) [ref4, ref5] relies on pre-trained weights of parametric softmax classifiers. This reveals again their post-hoc explainable nature. More critically, that also means [ref4, ref5] even cannot work without the initialization of a pre-trained softmax classifier. In sharp contrast, DNC is trained on ImageNet [9] from scratch. iii) [ref4, ref5] bring significant architectural change to the base classification network, while DNC is much more elegant and general -- just removing the softmax layer; iv) DNC evens defeats the *de facto*, softmax classifier on ImageNet and can be used as a backbone for downstream task (e.g., semantic segmentation). As far as we know, there is NO previous concept/prototype based explainable network (including [ref4, ref5]) can make this. We cite [ref4] as [81] in the new version ([ref5] had been cited as [72]).
>
> As for [ref1, ref2, ref3], please see **Clarification**. Thanks.
>
> [ref4] Neural Prototype Trees for Interpretable Fine-grained Image Recognition, CVPR 2021
>
> [ref5] This looks like that: Deep learning for interpretable image recognition, NIPS 2019

---

> ### Author Response · Authors · 2022-11-18
> **Clarification on Misunderstanding**
>
> Before providing point-to-point responses, we would like to first respectfully clarify a misunderstanding between the concepts of distance/similarity learning and distance-/similarity-based classification. Throughout the review, the reviewer repeatedly mentions metric learning methods [ref1-ref3]:
>
> - "In these works, learning and optimization of parameters is done on the basis of similarity/metric learning, ... "
>
> - "methods based on latent similarity metric learning ..., as the principle of classification/prediction for those methods is the same of closest-prototype picking."
>
> - "other approaches ... (e.g. supervised contrastive learning, circle loss, arcloss) that are not considered ..."
>
> - "This work seems to relate with metric learning, e.g. contrastive learning, approaches but the authors do not cover this part. Indeed, equation 7 reminds very clearly the 'contrastive loss function' (with the difference that only the min distance is considered in (7)), see [3] equation 1."
>
> [ref1] Circle Loss: A Unified Perspective of Pair Similarity Optimization, CVPR 2020
>
> [ref2] Arc Loss: Softmax with Additive Angular Margin for Answer Retrieval, AIRS 2019
>
> [ref3] Supervised Contrastive Learning, NIPS 2020
>
>
> -------------------------------------------------------------------------------------------
>
> **Clarification:** [ref1-ref3] focus on **distance/similarity learning**, while our DNC is a **distance-/similarity-based classifier**. [ref1-ref3] only use metric learning to better shape the representation space of softmax classifiers. Their trained models are STILL black-box, parametric classifiers. They are NOT distance-/similarity-based classifiers; they are just metric learning boosted parametric classifiers. They CANNOT provide any interpretability. Our DNC is a distance-based classifier. That is to say, its *classification decision mode* is based on distance/similarity (Eq. 7): given a test data sample, DNC directly classifies it to the class with the closest center, WITHOUT using parametric softmax. Learning such a distance-based classification network is clearly much more challenging.
>
> [ref1-ref3] are only to optimize the feature space by adjusting the arrangement between data samples, using, for example, contrastive loss. However, this is NOT the classification mode. They CANNOT use the contrastive loss or other metric learning losses to perform classification; they still use parametric softmax for classification. Differently, our DNC directly uses Eq. 7 for classification, and the cross-entropy loss can be directly applied to Eq. 7 (then it becomes a kind of metric loss). The principles of classification/prediction are TOTALLY different between DNC and these metric-learning methods.
>
> Distance-based classifiers rely on the similarity between samples and class representatives for classification (like our Eq. 7). Thus they naturally involve metric learning to get a better similarity measurement. There is no surprise that our Eq. 7 looks like some metric learning losses (every distance-based classifier is formulated in such a similarity form). But metric learning is not equivalent to distance-based classification. Distance-based classifiers also need to consider how to find/update reliable class representatives. Historically, metric learning and class center discovery are two key topics in the field of distance-/prototype-based classification and this is one of the motivations behind the proposal of metric learning. Please see [59] and our newly added Sec. N for more details. In page 5, we even had explicitly stated that "with such a nonparametric, distance-based scheme, DNC builds a closer link to metric learning [87–90]; DNC can even be viewed as learning a metric function to compare data samples, under the guidance of the corresponding semantic labels".
>
> To better address your concern and avoid potential misunderstandings, we review representative literature on metric/distance learning and discuss our relation to these efforts in Appendix Sec. N. We also address that "considering the similarity-/distance-based nature of DNC, the incorporation of metric learning based training objectives is also another promising direction for further boosting the performance." in Appendix Sec. O. We sincerely hope that the reviewer can better situate our work (as well as distance-based classification) in relation to the literature on distance learning  [ref1-ref3] and reconsider our contribution in light of the aforementioned context. Thanks.

---

> > ### Comment · Reviewer_1ipB · 2022-12-09
> > **thanks for the extended clarifications - discussion**
> >
> > Thanks for the preliminary overall clarifications of the doubts I expressed.
> >
> > It is clear to me that DNC learns both a distance metric and a distance-based classifier.
> > In my understanding, the main difference with other metric learning-based approaches is in the classification strategy, namely distance-based (with prototypes) and cross-entropy-based. Still, with e.g. arcloss (arcface) the resulting representations of classes (i.e. face identities) can be represented with central feature vectors, which might be seen as identity/class prototypes. In plain contrastive learning methods, the prototypes might also be obtained as center vector of clusters. This is however an indirect method to obtain prototypes, while in the DNC method finding the prototypes is explicitly formulated as part of the learning problem, which is per se an interesting contribution.
> >
> > I appreciate that the authors clarified this link with metric learning methods more explicitly, as it seems to me that DNC is very related and the work deserves proper contextualization.
> >
> > Please confirm my understanding, or further argue on an eventual misunderstanding.

---

> > > ### Author Response · Authors · 2022-12-09
> > > **Thanks for your response and further discussion**
> > >
> > > Thank you so much for your response.
> > >
> > > Yes, as you noticed, DNC automatically discovers class prototypes during network training and simultaneously learns the classifier on these prototypes from scratch. This is much more challenging and appealing.
> > >
> > >
> > > Moreover, we would like to reiterate three points.
> > >
> > >
> > > - First, in DNC, the prototypes have clear geometric meanings -- they are the centers of data samples in the embedding space or even can be representative data samples themselves. Such nonparametric and case-based reasoning properties are the source of DNC's similarity and interpretability and also greatly differentiate DNC from previous **black-box parametric** classifiers (like softmax classifier and arcloss) which treat class "prototypes" as fully learnable parameters.
> > >
> > > - Second, for previous metric learning-based approaches, one may obtain the prototypes as the center vectors of clusters after network training. However, this is meaningless, because such setting is much more simple and, undoubtedly, will lead to great performance drop (as they do not train on this) -- even for every deep network classifier, we can make clustering after network training, and use the cluster center for classification. Instead, DNC end-to-end learns a transparent classifier that can explicitly leverage ``quintessential'' past observations as evidence for classification.
> > >
> > >
> > > - Third, in principle, previous metric-learning algorithms can be integrated into DNC, but they cannot handle our setting -- end-to-end training a metric-based, nonparametric classifier.
> > >
> > > In short, DNC is essentially a new type of neural network classifier and is fundamentally different from the current de facto, parametric softmax-based regime.
> > >
> > > Finally, we would like to thank you again for your valuable comments, which help us better position our work in comparison to previous efforts.

---

> ### Author Response · Authors · 2022-12-06
> **Call for open dialogue**
>
> Dear Reviewer 1ipB,
>
> We first want to thank you again for reviewing our work. We understand you may have multiple responsibilities in your academic and life roles and your agenda may be preoccupied. But we would appreciate your valuable time to respond to our feedbacks based on your comments. We strongly feel that we have appropriately addressed all your concerns. Please let us know if you have any other concerns.
>
> Thank you.
>
> Authors.

---

### Author Response · Authors · 2022-11-18
**Summary of Revisions**

To all reviewers:

Thank you so much for your careful review and suggestive comments. We have revised our paper according to your comments (where the revised parts are highlighted in red color). The major changes are as follows:

1. We add experiments in comparison to distance (learning) based classifiers according to Reviewer 1ipB's comments in Appendix Sec. I, Table 18&19&20.
2. We add experiments on ImageNetv2 test sets according to Reviewer GtUv's comments in Appendix Sec. G, Table 16.
3. We add experiments on exploring transferability for the image classification task, using Caltech-UCSD Birds-200-2011 dataset, according to Reviewer GtUv's comments in Appendix Sec. F, Table 15.
4. We add experiments for comparing $k$-means $vs$ Sinkhorn-Knopp clustering algorithms, according to the comments of Reviewer 1ipB, GtUv, and k9FP in Appendix Sec. H, Table 17.
5. We add ablative experiments on the temperature parameter $\epsilon$ in Eq.6, according to Reviewer GtUv's comments in Appendix Sec. J, Table 22.
6. We study the performance when varying the number of centroids $K$ with the number of training samples of different classes, according to Reviewer 3KaQ's comments in Appendix Sec. J, Table 23.
7. We offer more detailed discussions regarding the GPU memory cost, according to Reviewer 3KaQ's comments in Appendix Sec. K, Table 24.
8. We highlight the use of second-order statistics and the automatic determination of the number of class centroids as a part of our future work according to Reviewer 3KaQ's comments in Appendix Sec. O.
9. We discuss our relation to distance/metric (learning) based classifiers according to Reviewer 1ipB's comments in Appendix Sec. N.
10. We discuss our relation to clustering based unsupervised learning algorithms according to Reviewer k9FP's comments in Sec. N.
11. We correct all typos and language in the current revision of the paper according to the comments of Reviewer GtUv and k9FP.

Sincerely yours,

Authors.

---

### Decision · Program_Chairs · 2023-01-20

**Decision:**

Accept: notable-top-25%

**Justification For Why Not Higher Score:**

combining clustering (centroids) in neural networks is not entirely new ideas, which can be traced back to early works, e.g. RBF

**Justification For Why Not Lower Score:**

The paper shows interesting results, e.g. semantic segmentation, and interpretability based on the sub-centroids.
Hence this will be useful to be highlighted to the ICLR researchers

**Metareview: Summary, Strengths And Weaknesses:**

This paper proposes deep nearest centroids, a new non-parametric classification layer. A number of sub-centroids are learned for each class, and the prediction is the class corresponding to the closest sub-centroid during testing. Good improvements are shown on CIFAR-10 (ResNet backbones) and ImageNet (ResNet + Swin backbones). Interesting results were also shown on semantic segmentation. In addition, the algorithm shows out-of-the-box interpretability, i.e. the sub-centroids can be used to explain classification decisions by the model.



**Note From Pc:**

if the above contains the word "oral" or "spotlight" please see: "oral" presentation means -> notable-top-5% and "spotlight" means -> notable-top-25%. As stated in our emails, we are disassociating presentation type from AC recommendations

**Summary Of Ac-Reviewer Meeting:**

NA as the only reviewer 1ipB who scores 5 has changed his mind to accept the paper